**The triple oxygen isotope composition of phytoliths as a proxy of continental atmospheric**
**humidity: insights from climate chamber and climate transect calibrations**
Anne Alexandre[1], Amaelle Landais[2], Christine Vallet-Coulomb[1], Clément Piel[3], Sébastien
Devidal[3], Sandrine Pauchet[1], Corinne Sonzogni[1], Martine Couapel[1], Marine Pasturel[1], Pauline
Cornuault[1], Jingming Xin[2], Jean-Charles Mazur[1], Frédéric Prié[2], Ilhem Bentaleb[4], Elizabeth
Webb[5], Françoise Chalié[1], Jacques Roy[3].
[1] CEREGE UM34, Aix-Marseille Université, CNRS, IRD, INRA, Aix en Provence, France
[2] Laboratoire des Sciences du Climat et de l'Environnement (LSCE/IPSL/CEA/CNRS/UVSQ),
Gif-sur-Yvette, France
[3]Ecotron Européen de Montpellier, UPS 3248, Centre National de la Recherche Scientifique
(CNRS), Campus Baillarguet, Montferrier-sur-Lez, France
[4] ISEM, Université de Montpellier, CNRS, IRD, EPHE, Montpellier, France
[5]Department of Earth Sciences, The University of Western Ontario, London, Ontario, Canada
Correspondance: alexandre@cerege.fr
**Abstract**
Continental atmospheric relative humidity (RH) is a key climate-parameter. Combined with
atmospheric temperature, it allows us to estimate the concentration of atmospheric water vapor
which is one of the main components of the global water cycle and the most important gas
contributing to the natural greenhouse effect. However, there is a lack of proxies suitable for
reconstructing, in a quantitative way, past changes of continental atmospheric humidity. This
reduces the possibility to make model-data comparisons necessary for the implementation of
climate models. Over the past 10 years, analytical developments have enabled a few laboratories
to reach sufficient precision for measuring the triple oxygen isotopes, expressed by the $^{17}$O-excess
($^{17}$O-excess = ln ($\delta^{17}$O + 1) – 0.528 x ln ($\delta^{18}$O + 1)), in water, water vapor and minerals. The $^{17}$O-
excess represents an alternative to deuterium-excess for investigating relative humidity conditions
that prevail during water evaporation. Phytoliths are micrometric amorphous silica particles that
form continuously in living plants. Phytolith morphological assemblages from soils and sediments
are commonly used as past vegetation and hydrous stress indicators. In the present study, we
examine whether changes in atmospheric RH imprint the $^{17}$O-excess of phytoliths in a measurable
way and whether this imprint offers a potential for reconstructing past RH. For that purpose, we
first monitored the $^{17}$O-excess evolution of soil water, grass leaf water and grass phytoliths in
response to changes in RH (from 40 to 100 %) in a growth chamber experiment where transpiration
reached a steady state. Decreasing RH from 80 to 40% decreases the $^{17}$O-excess of phytoliths by
4.1 per meg / % as a result of kinetic fractionation of the leaf water subject to evaporation. In order
to model with accuracy the triple oxygen isotope fractionation in play in plant water and in
phytoliths we recommend direct and continuous measurements of the triple isotope composition
of water vapor. Then, we measured the $^{17}$O-excess of 57 phytolith assemblages collected from top
soils along a RH and vegetation transect in inter-tropical West and Central Africa. Although
scattered, the [17]O-excess of phytoliths decreases with RH by 3.4 per meg / %. The similarity of the
trends observed in the growth chamber and nature supports that RH is an important control of [17]O-
excess of phytoliths in the natural environment. However, other parameters such as changes in the
triple isotope composition of the soil water or phytolith origin in the plant may come into play.
Assessment of these parameters through additional growth chambers experiments and field
campaigns will bring us closer to an accurate proxy of changes in relative humidity.

## 1    Introduction

Continental atmospheric relative humidity (RH) is a key climate-parameter. Combined with
atmospheric temperature, it allows scientists to estimate the concentration of atmospheric water
vapor which is one of the main components of the global water cycle and the most important gas
contributing to the natural greenhouse effect (e.g. Held and Soden, 2000; Dessler and Davis, 2010;
Chung et al., 2014). However, global climate models (GCMs) have difficulties to properly capture
continental humidity conditions (Sherwood et al., 2010; Risi et al., 2012; Fischer and Knutti,
2013). Although tropospheric RH results from a subtle balance between different processes
(including air mass origins and trajectories, large scale radiative subsidence, evaporation of falling
precipitation, detrainment of convective system, evapotranspiration), it is usually depicted as
rather constant in GCMs in agreement with thermodynamic coupling between atmospheric water
vapor and sea surface temperature (Bony et al., 2006; Stevens et al., 2017). A model-data
comparison approach is thus essential to progress on this issue. This approach has to be applicable
beyond the instrumental period to make use of past changes in atmospheric water vapor conditions.
There are multiple ways to reconstruct past continental temperature and precipitation, for instance
from pollen (Bartlein et al., 2010; Herbert and Harrison, 2016; Wahl et al., 2012) or tree ring data
(Labuhn et al., 2016; Lavergne et al., 2017). However, there is a serious lack of proxies suitable
for reconstructing, in a quantitative way, past variations in continental atmospheric RH. Indeed,
the stable isotopes of oxygen and hydrogen ($\delta^{18}O$ and $\delta D$) of tree rings can be influenced by several
parameters other than humidity (precipitation source, temperature). This limits the interpretation
of tree ring isotope series in terms of humidity changes to places where variations of these other
parameters are well constrained (Grießinger et al., 2016; Wernicke et al., 2015). A promising
method relies on the $\delta^{18}O$ and $\delta D$ of plant biomarkers (e.g. n-alkanes and fatty acids from leaf
waxes) recovered from soils (or buried soils) and sediments. It allows for an estimate in changes
in plant water deuterium-excess (d-excess = $\delta D$ - 8.0 x $\delta^{18}O$), linked to changes in precipitation
sources and RH. This method under development can however be biased by factors other than
climatic such as plant functional types and selective degradation of the biomarkers (e.g. Rach et
al., 2017; Schwab et al., 2015; Tuthorn et al., 2015).
Phytoliths are micrometric amorphous silica ($SiO_2$, $nH_2O$) particles that form continuously in
living plants. Silicon is actively absorbed by the roots (Ma and Yamaji, 2006) and is translocated
in the plant tissues where it polymerizes inside the cells, in the cell walls and in extracellular spaces
of stems and leaves. Silica polymerization appears to be an active physiological process, which
does not only depends on transpiration (Kumar et al., 2017). In grasses, which are well known
silica accumulators, silica accounts for several % of dry weight (d.w.) and is mainly located in the
stem and leaf epidermis. Phytolith morphological assemblages from soils and sediments are
commonly used as past vegetation and hydrous stress indicators (e.g. Aleman et al., 2012;
Backwell et al., 2014; Bremond et al., 2005a, 2005b; Contreras et al., 2014; Nogué et al., 2017 ;
Piperno, 2006). The potential of the $\delta^{18}O$ signature of phytoliths ($\delta^{18}O_{Phyto}$) from grasses for
paleoclimate reconstruction has been investigated through growth chamber and North American
Great Plains calibrations. It has been shown that the $\delta^{18}O_{Phyto}$ of grass stems weakly affected by
transpiration correlated with the $\delta^{18}O$ signature of soil water ($\delta^{18}O_{SW}$) and the atmospheric
temperature, as expected for a polymerization of silica in isotope equilibrium with the plant water
(Webb and Longstaffe, 2000, 2002, 2003, 2006). It has also been shown that $\delta^{18}O_{Phyto}$ from grass
leaves correlated with RH as expected for an evaporative kinetic isotope enrichment of the leaf
water (e.g. Cernusak et al., 2016) imprinted on $\delta^{18}O_{Phyto}$. However, because grass stem and leaf
phytoliths have the same morphology and are mixed in soil and sedimentary samples, these
calibrations were not sufficient for using $\delta^{18}O_{Phyto}$ of grassland phytolith assemblages as a
paleoclimatic signal. In tropical trees, silica is found in leaves, bark and wood and accounts for a
few % d.w. (e.g. Collura and Neumann, 2017). In the wood, silica polymerizes in the secondary
xylem supposedly unaffected by transpiration, in the form of Globular granulate phytolith types
(Madella et al., 2005; Scurfield et al., 1974; Welle, 1976). These phytoliths make up more than
80% of tropical humid forest and rainforest phytolith assemblages found in soils and sediments
(Alexandre et al., 2013; Collura and Neumann, 2017; Scurfield et al., 1974; Welle, 1976).
Examination of the $\delta^{18}O_{Phyto}$ of rainforest assemblages showed correlations with the $\delta^{18}O$ of
precipitation ($\delta^{18}O_{Pre}$) and the atmospheric temperature (Alexandre et al., 2012). However, in this
case, the use of $\delta^{18}O_{Phyto}$ did not further develop because it was applicable only to forested areas
and humid climatic periods, which is a major drawback for paleoclimatic reconstructions.
The triple isotope composition of oxygen in the water molecule represents an alternative for
investigating RH conditions prevailing during water evaporation. In the triple isotope system, the
mass-dependent fractionation factors between A and B ($^{17}\alpha_{A-B}$ and $^{18}\alpha_{A-B}$) are related by the
exponent $\theta_{A-B}$ ($^{17}\alpha_{A-B} = {}^{18}\alpha_{A-B}{}^{\theta}$ or $\theta_{A-B} = \ln{}^{17}\alpha_{A-B} / \ln{}^{18}\alpha_{A-B}$). The exponent can also be expressed
as $\theta_{A-B} = \Delta'^{17}O_{A-B} / \Delta'^{18}O_{A-B}$ with $\Delta'^{17}O_{A-B} = \delta'^{17}O_A - \delta'^{17}O_B$, $\Delta'^{18}O_{A-B} = \delta'^{18}O_A - \delta'^{18}O_B$, $\delta'^{17}O = \ln$
($\delta^{17}O + 1$) and $\delta'^{18}O = \ln (\delta^{18}O + 1)$. In the $\delta'^{17}O$ vs $\delta'^{18}O$ space, $\lambda_{A-B}$ represents the slope of the
data alignment during a mass-dependent fractionation process between A and B. $\lambda_{A-B}$ is an
empirical way to assess $\theta_{A-B}$ (Li et al., 2017). It has been recently estimated that $\theta$ equals 0.529
for liquid-vapor equilibrium ($\theta_{equil}$; Barkan and Luz, 2005) and 0.518 for vapor diffusion in air
(Barkan and Luz, 2007). It has additionally been shown that meteoric waters plot along a line with
a slope $\lambda$ of $0.528 \pm 0.001$. The departure from the meteoric water line is conventionally called
$^{17}O$-excess ($^{17}O$-excess $= \delta'^{17}O - 0.528 \times \delta'^{18}O$) (Luz and Barkan, 2010). In case of mass-dependent
fractionation processes, the magnitudes of the $^{17}$O-excess in waters and minerals are very small
and measurement of the $^{17}$O-excess, expressed in per meg ($10^{-3}$‰) *vs* VSMOW, requires very high
analytical precisions.
In the water cycle, the $^{17}$O-excess variations mainly result from diffusion processes, while
equilibrium fractionation does not lead to important departure from the meteoric water line.
Theoretical and empirical estimations have shown that in contrast to d-excess, and except at very
high latitudes, changes in water $^{17}$O-excess are not significantly impacted by temperature (~0.1 per
meg / °C ; Uemura et al., 2010) and much less sensitive to distillation processes (Angert et al.,
2004; Barkan and Luz, 2007; Landais et al., 2008; Uemura et al., 2010; Steig et al., 2014). Changes
in water $^{17}$O-excess are thus essentially controlled by evaporative kinetic fractionation. The $^{17}$O-
excess decreases in the evaporating water and increases in the vapor phase when RH decreases at
evaporative sites (e.g. sea surface, lake surface, soil surface or leaf surface).  Over the last ten
years, a few studies used the $^{17}$O-excess of water to interpret ice core archives in climatic terms
(Guillevic et al., 2014, Schoeneman et al., 2014; Winkler et al., 2012; Landais et al., 2008, 2012).
They supported that $^{17}$O-excess is a marker of RH, sea-ice extent at the moisture source, and air
mass mixing (Risi et al., 2010) except at the very high latitudes of East Antarctica where
temperature can have a significant influence. The observed variations of $^{17}$O-excess in Greenland
ice cores of ~20 per meg maximum were thus interpreted as variations of RH or sea-ice extent at
the source region and coincide with variations in the low to mid latitude water cycle as recorded
by other proxies (such as $CH_4$ or $\delta D$ of $CH_4$) (Guillevic et al., 2014). An even smaller number of
studies measured or attempted to model the $^{17}$O-excess of rainwater at low and temperate latitudes
(Affolter et al., 2015; Landais et al., 2010b; Li et al., 2015; Luz and Barkan, 2010; Risi et al.,
2013). The observed variations in $^{17}$O-excess, partly explained by convective processes and re-
evaporation of precipitation, were of the order of 30-40 per meg, either during a rainy event or
along climatic gradients. Only two studies focused on open surface waters, and showed that
variations of the $^{17}$O-excess ranged from tens to hundreds of per meg when the surface water
underwent strong evaporative enrichment (Surma et al., 2015; Luz and Barkan, 2010), in
agreement with the Craig and Gordon (1965) formulation. The most important variations in $^{17}$O-
excess occur at the plant-atmosphere interface. In leaf water, variations higher than 200 per meg
were encountered (Landais et al., 2006; Li et al., 2017). Difference in $^{17}$O-excess between leaf
water subject to evaporation (LW) and stem water (SW) not subject to evaporation, increased with
decreasing RH (from 100 to 30 %), as expected for processes dominated by kinetic fractionation.
When measuring a sequence of LW- SW couples sampled under different climatic conditions, the
slope of the line linking their triple isotope composition and named $\lambda_{transp}$, equivalent to $\theta_{LW-SW}$, was
found to change with RH. This pattern was neither influenced by the plant species nor by the
environmental conditions (e.g. atmospheric temperature, soil water conditions) (Landais et al.,
2006). However opposite trends of $\lambda_{transp}$ with RH were observed from one study to another
(Landais et al., 2006; Li et al., 2017). This discrepancy was attributed to the possibility that steady
state is not always reached during sampling and to likely differences in isotope composition of the
ambient vapor, a parameter of the Craig and Gordon model that is often not measured but estimated
(Li et al., 2017).
While $^{17}O$-excess measurements of waters were expanding, analyses of the triple oxygen isotope
composition of minerals (mostly silicates and carbonates) were also developed, allowing estimate
of fractionation during polymerization and providing constraints on both temperature and isotope
composition of the water source (Pack and Herwartz, 2014; Levin et al., 2014; Passey et al., 2014;
Herwartz et al., 2015; Miller et al., 2015; Sharp et al., 2016). Variations of $^{17}O$-excess of the order
of tens to hundreds of per meg were reported from one mineral to another. For most of the studies
cited above, the objective was to discriminate between high and low temperature formation
processes or to decipher from which type of water the mineral formed (i.e. sea water, hydrothermal
water, meteoric or surface water). The $^{17}O$-excess of biogenic and sedimentary carbonates was
also investigated as a potential record of evaporating water sources (Passey et al., 2014).With
regard to silicate-water fractionation, the relationship between the three oxygen isotopes defined
by $\theta_{SiO2-water}$ was estimated between 0.521 and 0.528, increasing logarithmically with temperature
(Sharp et al., 2016).
In the present study, in the light of the recent findings cited above, we examined whether changes
in atmospheric RH imprint the $^{17}O$-excess of phytoliths ($^{17}O$-excess$_{Phyto}$) in a measurable way and
whether this imprint offers a potential for reconstructing past RH. For that purpose, we first
monitored the $^{17}O$-excess evolution of soil water, grass leaf water and grass phytoliths in response
to changes in RH in a growth chamber experiment. Then, we measured the $^{17}O$-excess$_{Phyto}$ from
57 phytolith assemblages collected in soil tops along a RH and vegetation transect in inter-tropical
West and Central Africa. Relationships between $^{17}O$-excess$_{Phyto}$ and RH were looked for and
assessed on the basis of previous quantifications of kinetic isotope enrichment of leaf water and
equilibrium fractionation between water and silica. Results from the natural sampling were
compared to the ones from the growth chamber experiment to evaluate the importance of RH in
controlling $^{17}O$-excess$_{Phyto}$ in natural environment.
**2      Materials and methods**
**2.1      Samples from the growth chamber experiment**
*Festuca arundinacea*, commonly referred to as tall fescue, is widely distributed globally as forage
and an invasive grass species (Gibson and Newman, 2001) and can adapt to a wide range of
conditions. In 2016, *F. arundinacea* (Callina RAGT Semences) was grown in three chambers
under three conditions of RH (ca. 40, 60 and 80 %) kept constant using wet air introduction and
ultrasonic humidifier. We checked that the humidifiers did not lead to any isotope fractionation
between the water in their reservoirs and the vapor delivered. Temperature and light intensity were
kept constant at 25 ± 0.6 (standard deviation (SD)) °C and 293 ± 14 (SD) mmol / m$^2$ / sec
respectively.
In a 35 L tank (53 x 35 x 22 cm), 20 kg of dried commercial potting soil were packed above a 1.6
cm layer of quartz gravel. A porous cup for water extraction was placed in the soil with its
extraction tube hermetically extending outside of the tank walls. The soil was irrigated with 10 L
of the same water as the one used for the humidifier. Four grams of seeds were sown along four
rows in each tank, resulting in about 6000 seedlings. Each tank was then placed in a chamber and
was irrigated from a Mariotte bottle (25 L) placed next to it. The Mariotte system was set so that
a water saturated level of 5 cm remained constant at the base of the tank. The irrigation water was
supplemented with 105 mg/L of $SiO_2$ (in the form of $SiO_2$ $K_2O$). Ten days after germination, agar-
agar (polysaccharide agarose) was spread on the soil surface around the seedlings (about 8 cm
tall), to prevent any evaporation (Alexandre et al., 2016).
A fourth tank was kept at 100% of RH thanks to the installation of a 20 cm high plexiglass cover,
in a forth chamber set at 80 % of RH. In this case no agar-agar was added and the vapor around *F.*
*arundinacea* came from evaporation and transpiration of the soil water. Otherwise the treatment
was the same as in the other chambers.
For each humidity condition, three to four harvests were made at intervals of 10-14 days. The 20-
25 cm long leaves were cut at two cm above the soil level and weighed. From the first to the fourth
sampling, the harvested wet leaves increased from 15-20 g (10 days of growth) to 40-60 g (14 days
of growth). Three to five g of leaves were put in glass gastight vials and kept frozen for bulk leaf
water extraction. The remaining leaves were dried for phytolith extraction. Forty mL of irrigation
water from the Mariotte bottle, and of soil water from the porous cup, were kept at 5°C before
analyses.
After each harvest, the tanks were left in their chamber of origin but the 40, 60 and 80 % RH
treatments were rotated between the growth chambers so that the four replicates of a given RH
treatment would come from at least two different chambers. The 100 % humidity was set up in a
unique chamber during the entire duration of the experiment. The harvested leaves in this treatment
were often covered by condensation drops which were blotted between two sheets of wiping paper,
rapidly after harvesting. The experimental setup details and the harvest list are given in table 1.

## 2.2     Samples from the natural climate transects

Fifty-seven top soil samples were collected during several field trips along vegetation and humidity
transects in Mauritania and Senegal (Bremond et al., 2005b ; Lézine, 1988; Pasturel, 2015)
(Lezine, 1988) Gabon (Lebamba et al., 2009) and Congo (Alexandre et al., 1997) in the saharan,
sahelian, sudanian, guinean and congolian bioclimatic zones, respectively (White et al., 1983).
Samplings, phytolith extractions and phytolith morphological assemblages descriptions are given
in the above-mentioned studies, except for the samples of Gabon from which phytoliths were
chemically treated and counted in the frame of the present study.
The sampled site location as well as the associated climatic and oxygen isotope variables are given
in Table 2. The vegetation overlying the sampled soils was categorized into savanna (Mauritania,
Senegal), wooded savanna (Senegal), humid forest (Gabon and Congo) and enclosed savanna
(Gabon). For each sampled site, yearly climate average were calculated from the monthly means
of temperature, precipitation, RH and diurnal temperature, extracted from the Climate Research
Unit (CRU) 1961 - 1990 time series (10' spatial resolution; http://www.cru.uea.ac.uk, Harris et al.,
2013, CRU 2.0). Mean Annual Precipitation (MAP), Mean Annual Temperature (MAT) and mean
annual RH range from 49 to 2148 mm, 24.3 to 29.8 °C and 40.2 to 82.5 %, respectively.  In
addition, in order to get a proxy of RH during wet months, likely those of the grass growing season,
averaged RH monthly means for months with at least one day with precipitation higher than 0.1
mm (RH-rd0>1) was calculated. It ranges from 56.3 to 82.5 %. As maximum transpiration is
supposed to be reached around 15:00 UTC we also calculated RH and RH-rd0>1 at 15:00 (RH15
and  RH15-rd0>1, respectively) according to New et al. (2002) and Kriticos et al. (2012).  For each
sampling site, estimates of  $\delta'^{18}O$ of precipitation for the months with at least one day with
precipitation higher than 0.1 mm ($\delta'^{18}O_{Pre-rd0>1}$) were calculated from $\delta^{18}O$ of precipitation
extracted from The Online Isotopes in Precipitation Calculator-version OIPC2-2
(http://www.waterisotopes.org; Bowen and Revenaugh, 2003; Bowen and Wilkinson, 2002;
Bowen et al., 2005) and weighted by the amount of precipitation. The estimates range from -1.51
to -4.46 ‰. There is currently no data on the $^{17}O$-excess of precipitation ($^{17}O$-excess$_{Pre}$) at these
sites.

## 2.3    Phytolith chemical extractions

Phytoliths from soils were extracted following Crespin et al. (2008) using HCl, $H_2O_2$, $C_6H_5Na_3O_7$
and $Na_2O_4S3-H_2O$ at 70 °C, and a $ZnBr_2$ heavy liquid separation. It has been shown that up to a
temperature of 70 °C the extraction has no effect on the $\delta^{18}O$ (Crespin et al., 2008). We verified
that it did not have any effect on the $^{17}O$-excess either, using our internal standard MSG extracted
at 60 and 70°C (Crespin et al., 2008). The obtained $^{17}O$-excess values were similar (-211 and -243
per meg, respectively) given our reproducibility of ±34 per meg (see section 2.6.1). Phytoliths
from *Festuca arundinaceae* were thus extracted using a high purity protocol with HCl, $H_2SO_4$,
$H_2O_2$, $HNO_3$, $KClO_3$ and KOH at 70 °C following Corbineau et al. (2013).

## 2.4    Phytolith counting

Phytolith assemblages from the humidity transects were mounted on microscope slides in Canada
Balsam, for counting, at a 600X magnification. More than 200 identifiable phytoliths with a
diameter greater than 5 μm and with a taxonomic significance were counted per sample. Three
repeated counting gave an error of ± 3.5 % (SD). Phytoliths were named using the International
Code for Phytolith Nomenclature 1.0 (Madella et al., 2005) and categorized as Globular granulate
type produced by the wood (Scurfield et al., 1974; Kondo et al., 1994), palm Globular echinate
type and grass types comprising Acicular, Bulliform, Elongate psilate, Elongate echinate,
Bulliform cells, and Grass Short Cells types. For each sample from the natural transects, the
phytolith index d/p, a proxy of tree cover density (Alexandre and Bremond, 2009; Bremond et al.,
2005a), was calculated. It is the ratio of Globular granular phytolith category (Madella et al., 2005)
formed in the secondary xylem of the dicotyledon (d) wood to the grass short cell phytolith
category formed in the epidermis of grasses or Poideae (p) (Collura and Neumann, 2017; Scurfield
et al., 1974; Welle, 1976). Those two categories make up most of the phytolith assemblages
recovered from inter-tropical soils ( Alexandre et al., 1997, 2013; Bremond et al., 2005b, 2005a).
Phytolith assemblages from the *F. arundinacea* samples were also mounted and counted. The
phytolith types were categorized according to their cell of origin in the epidermis into Epidermal
short cell, Epidermal long cell, Bulliform cell and Hair acicular.

## 274 2.5 Leaf and soil water extraction

Leaf water was extracted using a distillation line. Leaves were introduced in a glass tube connected
to the distillation line, and frozen through immersion of the glass tube in liquid nitrogen. While
keeping the sample frozen, the distillation line was pumped to reach a vacuum higher than $5.10^{-2}$
mbar. The pumping system was then isolated and the glass sample tube warmed to 80°C.
Meanwhile, at the other end of the distillation line, a glass collecting tube was immersed in liquid
nitrogen to trap the extracted water. To avoid condensation, the line between the sample tube and
the collection tube was heated with a heating wire. The distillation was completed after six hours.
In order to remove volatiles from the extracted water, a few granules of activated charcoal were
added and the water slowly stirred for 12 h.
Soil water was extracted using a 31mm porous ceramic cup. Brown or yellow-colored samples
were filtered at 0.22μm, but remained colored after filtration, indicating the presence of soluble
compounds.

## 287 2.6 Isotope analyses

The oxygen isotope results are expressed in the standard δ-notation relative to VSMOW.

### 289 2.6.1 Phytoliths

Phytolith samples of 1.6 mg were dehydrated and dehydroxylated under a flow of $N_2$ (Chapligin
et al., 2010) and oxygen extraction was performed using the IR Laser-Heating Fluorination
Technique at CEREGE (Aix-en-Provence, France) (Alexandre et al., 2006, Crespin et al., 2008;
Suavet et al., 2010). The purified oxygen gas ($O_2$) was passed through a −114 °C slush to refreeze
gases interfering with the mass 33 (e.g. NF), potentially produced during the fluorination of
residual organic N, before being sent to the dual-inlet mass spectrometer (ThermoQuest Finnigan
Delta Plus). The composition of the reference gas was determined through the analyses of NBS28
for which isotope composition has been set to $\delta^{18}O$=9.60 ‰, $\delta^{17}O$=4.99 ‰ and $^{17}O$-excess = -65
per meg. During the measurement period, reproducibility (SD) of the analyses of the working
quartz standard (Boulangé 2008) against which the isotope composition of the sample gas was
corrected on a daily basis (3 quartz standards were analysed per day) was ± 0.20 ‰, ± 0.11 ‰ and
± 22 per meg for $\delta^{18}O$, $\delta^{17}O$ and $^{17}O$-excess respectively (n = 63; one run of eight dual inlet
measurements). For every session of measurement, the effectiveness of the entire dehydration and
IR-Laser-Fluorination-IRMS procedure was checked through the analysis of a working phytolith
standard (MSG60) with $\delta^{18}O = 36.90 \pm 0.78$ ‰, $\delta^{17}O = 19.10 \pm 0.40$ ‰ and $^{17}O$-excess = -215 ±
34 per meg (n = 29). For comparison, the inter-laboratory pooled value for MSG60 is $\delta^{18}O = 37.0$
± 0.8 ‰ (Chapligin et al., 2011). Recent measurements of the silicate reference materials UWG-2
garnet (Valley et al., 1995) and San Carlos (SC) olivine gave the following values: $\delta^{18}O_{UWG-2} =$
$5.72 \pm 0.12$ ‰, $\delta^{17}O_{UWG-2} = 2.95 \pm 0.06$ ‰, $^{17}O$-excess $_{UWG-2} = -68 \pm 27$ per meg (n = 5), $\delta^{18}O_{SC} =$
$4.95 \pm 0.22$ ‰, $\delta^{17}O_{SC} = 2.56 \pm 0.12$ ‰, $^{17}O\text{-excess}_{SC} = -49 \pm 24$ per meg (n = 3). For comparison,
silicate analyses presented in Sharp et al. (2016) are normalized to a $\delta^{18}O$ value for San Carlos
Olivine of 5.3 ‰ and a $^{17}O$-excess value of -54 per meg. As previously discussed in Suavet et al.
(2010), a large scatter is often observed for SC olivine $\delta^{18}O$ and $\delta^{17}O$ values measured in a given
laboratory or from a laboratory to another. This is probably attributable to the heterogeneity of the
analyzed samples. At CEREGE, the internal standard of SC olivine is prepared from a number of
millimetric crystals with possibly different oxygen isotope composition. The $\delta^{18}O$ and $\delta^{17}O$ values
from Suavet et al. (2010), Tanaka and Nakamura (2013) Pack et al. (2016), Sharp et al. (2016) and
the present study average $5.29 \pm 0.23$ (1 SD) ‰ and $2.72 \pm 0.12$ (1 SD) ‰, respectively.
Nevertheless, despite the large SD on $^{18}O$ and $\delta^{17}O$ measurements, the SC olivine $^{17}O$-excess
appears relatively constant ($-71 \pm 23$ (1 SD)) per meg.

## 2.6.2   Leaf water

Leaf water was analyzed at LSCE (Gif sur Yvette, France) following the procedure previously
detailed in Landais et al. (2006). In summary, a fluorination line was used to convert water to
oxygen using $CoF_3$ heated at 370°C in a helium flow. The oxygen was then trapped in a tube
immersed in liquid helium before being analyzed by dual inlet IRMS (ThermoQuest Finnigan
MAT 253 mass spectrometer) against a reference oxygen gas. All measurements were run against
a working $O_2$ standard calibrated against VSMOW. The resulting precisions (2 runs of 24 dual
inlet measurements) were 0.015 ‰ for $\delta^{17}O$, 0.010 ‰ for and $\delta^{18}O$ and 5 per meg for $^{17}O$-excess.

## 2.6.3   Irrigation and soil waters

Irrigation and soil water were analyzed at the Ecotron of Montpellier (France) with an isotope laser
analyzer (Picarro L2140i) operated in $^{17}O$-excess mode using an auto-sampler and a high precision
vaporizer. Each water sample was used to fill three vials randomly dispatched in four groups of
six samples (three replicates per sample). Each sample group was bracketed by three working
standards (Giens-1, Iceberg-1 and Eco-1). Ten injections were performed for each vial, and the
results of the first six injections were discarded to account for memory effects. Following IAEA
recommendations (IAEA, 2013), each liquid measurement sequence was started with two vials of
deionized water for instrument conditioning.
The isotope compositions of each sample group were calibrated using the three interpolated mean
values obtained for the bracketing working standards (Delattre et al., 2015). All isotope ratios were
normalized on the VSMOW2/SLAP2 scale, with an assigned SLAP2 $^{17}O$-excess value of zero,
following the recommendations of Schoenemann et al. (2013). The resulting precisions (3
replicates) were 0.02 ‰, 0.01 ‰ and 10 per meg for $\delta^{17}O$, $\delta^{18}O$ and $^{17}O$-excess (n=31).
The three working standards were also analyzed using the fluorination/IRMS technique used for
leaf water analyses at LSCE. The $^{17}O$-excess maximum difference was 6.4 per meg, which is lower
than the analytical precision obtained using the laser spectrometer (Table S1a).
In order to assess that soluble organic compounds present in some soil water samples did not
impact the laser analyzer isotope measurements (Martín-Gómez et al., 2015), a representative set
of colored samples were analyzed with and without the Picarro micro combustion module (MCM)
set up between the high precision vaporizer and the analyzer inlet. This system was designed to
partly remove organic volatile compounds using a catalytic process. The obtained isotope
compositions were not significantly different (Table S1b), suggesting that organic compounds
were either in low concentration, and/or did not interfere in the spectral window used by the
analyzer. Therefore, the other soil water samples were analyzed without the MCM.
**3    Results**
**3.1    Growth chamber experiment**
$\delta'^{18}O$ and $^{17}O$-excess of the irrigation water (respectively $\delta'^{18}O_{IW}$ and $^{17}O$-excess$_{IW}$) average –5.59
$\pm$ 0.006‰ and 26 $\pm$ 5 per meg, respectively. $\delta'^{18}O$ and $^{17}O$-excess of the soil water (respectively
$\delta'^{18}O_{SW}$ and $^{17}O$-excess$_{SW}$) average -2.89 $\pm$ 0.19 ‰ and 16 $\pm$ 8 per meg, respectively (table S2).
The isotope difference is thus significant for $\delta'^{18}O$, less significant for $^{17}O$-excess, according to
the analytical error. Although evaporative kinetic fractionation of the top soil water suctioned by
the porous cup under vacuum cannot be ruled out, isotopic exchanges between the soil water and
oxygen-bearing phases of the rhizosphere may also have impacted the soil water isotopic
composition (Bowling et al., 2017; Chen et al., 2016; Oerter et al., 2014; Orlowski et al., 2016).
Hereinafter, we consider the isotope signatures of the water absorbed by the roots of *F.*
*arundinacea* to be equivalent to the irrigation water that fed the saturation level at the base of the
tank. This water was reached by the deepest roots, as observed on a cross-section of the soil after
the end of the experiment, and likely reached the upper roots by capillarity.
The transpiration of *F. arundinacea* increases rapidly from 0.03 to 0.6 L / day from 100 to 60 %
RH and more slowly from 60 to 40 % RH where it reaches 0.61 L / day (averages of the replicates,
Table 1). In response to decreasing RH, $\delta'^{18}O$ (table S2) and $^{17}O$-excess (fig. 1a) values of the bulk
leaf water ($\delta'^{18}O_{LW}$ and $^{17}O$-excess$_{LW}$) show clear increasing and decreasing trends, respectively.
The averaged $^{18}O$-enrichment of bulk leaf water relatively to irrigation water ($\Delta'^{18}O_{LW-IW}$)
increases from 100 to 60 % of RH and seems to be stabilizing from 60 to 40 % RH (fig. 1b; Table
1). For 100 % RH, the high standard deviations (SD) associated with $\delta^{18}O_{LW}$ (table S2), and
consequently with $\Delta'^{18}O_{LW-IW}$ (Table 1), are due to the very high $\delta^{18}O_{LW}$ value of sample P3-100-
10-05-16. However, as we do not have any explanation for this high value, this data was not
excluded from further calculation. The $^{17}O$-excess values associated with the enrichment $\Delta'^{18}O_{LW-}$
$_{IW}$ (or $^{17}O$-excess$_{e\ LW-IW}$ = $\Delta'^{17}O_{LW-IW}$ - 0.528 x $\Delta'^{18}O_{LW-IW}$) are scattered for a given RH. The
averaged value however follows a clear pattern (fig. 1c; table 1): it decreases slowly from 100 to
80 % RH (from -88 $\pm$ 48 to -75 $\pm$ 20 per meg,) and more rapidly from 80 to 40% RH where it
reaches -159 $\pm$ 9 per meg. When the relationship is linearized, the slope of the line between $^{17}O$-
excess$_{e\ LW-IW}$ and 40 to 80 % RH is 2.3 per meg/% (fig. 1f). The raw values of $\theta_{LW-IW}$ do not show
any significant trend with RH and average 0.519 $\pm$ 0.002. The slope $\lambda_{LW-IW}$ of the line linking
$\Delta'^{17}O_{LW-IW}$ and $\Delta'^{18}O_{LW-IW}$ (table 1) is 0.518.
The average phytolith content ranges from 1.1 to 0.1% d.w. Silicification of the leaf blade of *F.*
*arundinacea* increases with increasing transpiration and decreasing humidity (Table 1). Phytolith
morphological identification shows that they formed preferentially in the epidermal short cell and
to a smaller extent in the epidermal long cells (fig. 2). The proportion of silicified long cells,
increases with increasing transpiration and decreasing RH (Table 1). Some hair and bulliform cells
were also silicified, but in much smaller quantities. $\delta'^{18}O$ and $^{17}O$-excess of phytoliths ($\delta'^{18}O_{Phyto}$
and $^{17}O$-excess$_{Phyto}$ respectively) show the same general trends with RH as $\delta'^{18}O_{LW}$ and $^{17}O$-
excess$_{LW}$ (fig. 1a, table S2).
The average value of the $^{18}O$-enrichment of phytoliths relative to the bulk leaf water ($\Delta'^{18}O_{Phyto-}$
$_{LW}$) increases slowly (from $27.97 \pm 6.97$ to $28.47 \pm 0.38$‰) when RH decreases from 100 to 80 %
and more rapidly from 80 to 40% where it reaches $32.32 \pm 1.92$ ‰ (fig. 1b, Table 1).  With regard
to the enrichment of phytoliths relative to the irrigation water, $\Delta'^{18}O_{Phyto-IW}$ shows the same trend
with RH as $\Delta'^{18}O_{LW-IW}$ (fig.1b, table 1). $^{17}O$-excess$_{Phyto}$ and $^{17}O$-excess$_{e\ Phyto-IW}$ shows the same
decreasing trend with RH as $^{17}O$-excess$_{e\ LW-IW}$ (fig. 1c, Table 1). When the relationships of $^{17}O$-
excess$_{Phyto}$ and $^{17}O$-excess$_{e\ Phyto-IW}$ with 40 to 80 % RH are linearized, the slopes of the lines are
4.1 and 4.3 per meg/%, respectively (fig. 1d, 1f). A Student's t-test (relevant when the variance of
two data sets are equal; Andrade and Estévez-Pérez, 2014), calculated on the $^{17}O$-excess$_{e\ LW-IW}$ *vs*
RH and $^{17}O$-excess$_{e\ Phyto-IW}$ *vs* RH data sets shows that the slopes of the lines are not statistically
different for a 75% confidence interval. Thus, the link between $^{17}O$-excess$_{e\ Phyto-IW}$ and RH is
mainly due to the leaf water $^{17}O$-excess dependency to RH. The raw values of $\theta_{Phyto-LW}$ appears
constant, averaging $0.52 \pm 0.001$ (table 1).
## 3.2    Natural samples
Values of $\delta'^{18}O_{Phyto}$ and $^{17}O$-excess$_{Phyto}$ range respectively from 23.79 to 38.16 ‰ and from -140
to -290 per meg (table 2). The variations are in the same order of magnitude as for the growth
chamber experiment. The estimates of $\delta^{18}O_{Pre}$ vary little along the sampled transect (from -4.46 to
−3.22 ‰). No relationship is observed between $\delta'^{18}O_{phyto}$ or the $^{18}O$-enrichment of phytoliths
relatively to precipitation ($\Delta'^{18}O_{Phyto-Pre}$) and MAP, MAT or RH (fig. 3, table 2).
Although scattered, the $^{17}O$-excess$_{Phyto}$ values show a significant positive linear correlation with
RH (fig. 4), regardless of which RH variable is taken into account. After excluding two outliers,
the slopes of the correlation lines are 2.1 and 2.2 when RH and RH15 are taken into account, 3.4
when either RH-rd0>1 or RH15-rd0>1 are considered. The relationship obtained between $^{17}O$-
excess$_{Phyto}$ and RH-rd0>1 (i.e. RH of the wet months during which plant grow) is the closest to the
one obtained between $^{17}O$-excess$_{phyto}$ and RH in the growth chambers (fig. 4b). It can be expressed
as follows (Eq.3):
$^{17}O$-excess$_{phyto}$ = 3.4 x (RH-rd0>1) - 460        ($r^2 = 0.48$; $p < 0.001$)        Eq. 1
where $^{17}O$-excess$_{phyto}$ is expressed in per meg *vs* VSMOW and RH in %.
The excluded outliers (Table 3) are RIM1 and C3L4. RIM1 presents a very low $^{17}O$-excess (-305
per meg) relative to the $^{17}O$-excess of the samples with close RH-rd0>1, i.e. from 71 to 74 %
(average of -237 ± 32 per meg for 82-78, 83-116 and 83-115). C3L4 is located next to C4L3 and
under similar averaged RH but presents a $^{17}$O-excess higher by 133 per meg. RIM1 and C3L4
show morphological patterns very similar to the other assemblages with the same range of RH.
Thus, the discrepancies may lie either in the fact that local RH variations may not be reflected in
RH averaged estimates for 10' ($\approx$ 185 km$^2$) or in the particularity of the isotope composition of the
local soil water (see discussion below).
The phytolith index d/p ranges from 0.01 to 0.08 in savanna, from 0.14 to 0.49 in wooded savanna,
from 0.76 to 1.58 in enclosed savanna and from 1.84 to 6.78 in humid forests (Table 2). This
unambiguous increase of d/p with tree cover density is in agreement with previous calibrations
performed for the West African area (Bremond et al., 2005b). Interestingly, under high RH
conditions, humid forest and enclosed savanna that are characterized by a large range of d/p
represent a small range of $^{17}$O-excess. Conversely, under lower RH conditions, savanna and
wooded savanna that are characterized by a small range of d/p represent a large range of $^{17}$O-excess
(fig.5). This absence of relationship between $^{17}$O-excess and tree cover density is also mirrored in
figure 4 where phytolith samples from different vegetation types (i.e. savanna *vs* wooded savanna
or humid forests *vs* enclosed savanna), that have developed under the same RH conditions, have
the same range of $^{17}$O-excess.
**4      Discussion**
**4.1      Imprint of changes in atmospheric RH on the $^{17}$O-excess of leaf water**
In the bulk leaf water, the trends observed between $\Delta'^{18}O_{LW-IW}$ or $^{17}$O-excess$_{e\,LW-IW}$ and RH are in
agreement with an evaporative kinetic fractionation that increases when RH decreases, as expected
from previous studies on the $\delta^{18}O$ or $^{17}$O-excess evolution of leaf water (e.g. Cernusak et al., 2016;
Landais et al., 2006; Li et al., 2017). The obtained values of $\theta_{LW-IW}$ average (0.519) and of $\lambda_{LW-IW}$
(0.518) are respectively close and similar to the value of $\theta_{diff}$ calculated for the diffusion of vapor
in air (0.518; Barkan and Luz, 2007). As schematically described in Landais et al. (2016), $\lambda_{transp}$
(equivalent to $\lambda_{LW-IW}$) represents the interplay among three processes in the leaf boundary layer:
1) the equilibrium fractionation, which is only temperature-dependent (Majoube, 1971) and drives
the isotope composition of leaf water along the equilibrium water line ($\theta_{equil}$ = 0.529); 2) the
diffusion transport leading to increasing kinetic fractionation with decreasing relative humidity
along the diffusion line; 3) the isotope exchange of leaf water with atmospheric water vapor,
decreasing from turbulent to laminar and molecular leaf boundary layer vapor transport conditions
(e.g. Buhay et al., 1996). In the case of the growth chamber experiment, the fact that $\theta_{LW-IW}$
and $\lambda_{LW-IW}$ are respectively close and similar to $\theta_{diff}$ supports that the increasing diffusion of vapor
in air when RH decreases or transpiration increases is the main process controlling the evolution of
$^{17}$O-excess$_{LW}$. At high humidity (80-100% RH), the kinetic fractionation likely reaches its
minimum as the diffusion process becomes limited.
The $\delta^{18}O_{LW}$ is commonly modelled as a function of the isotope composition of absorbed water, the
isotope composition of water vapor, and RH (Craig and Gordon, 1965). The Craig and Gordon
simple approach overestimates $\delta^{18}O_{LW}$ and different corrections have been proposed to take into
account the diffusion of the evaporating water back to the leaf lamina and the advection of less
evaporated stem water (i.e. the Péclet effect, Buhay et al., 1996; Helliker and Ehleringer, 2000;
Roden et al., 2000; Farquhar and Gan, 2003; Farquhar and Cernusak, 2005; Ripullone et al., 2008;
Treydte et al., 2014). In the growth chamber experiment, where water availability, relative
humidity, and temperature were kept constant, we assume that transpiration rapidly reached a
steady state and that the isotope composition of transpired water was the same as that of the
irrigation water entering the plant (e.g. Welp et al., 2008). A tentative estimate of the theoretical
value of $\Delta'^{18}O_{LW-IW}$, $\Delta'^{17}O_{LW-IW}$ and $^{17}O\text{-excess}_{e\ LW-IW}$ was performed using the equations proposed
for $^{18}O$-enrichment by Cernusak et al. (2016) (table S3). For calculating the $\Delta^{17}_{LW-IW}$ we used for
the equilibrium and kinetic fractionations (respectively $^{17}\alpha_{eq}$ and $^{17}\alpha_k$ in table S3) $^{17}\alpha_{eq} = {}^{18}\alpha_{eq}{}^{0.529}$
and $^{17}\alpha_k = {}^{18}\alpha_k{}^{0.518}$. As expected, the predicted $\Delta'^{18}O_{LW-IW}$ values were all higher than the observed
values by several ‰. Helliker and Ehleringer (2000) proposed, for monocotyledonous species
characterized by a vertical parallel veinal structure, to use instead of the Craig and Gordon model
the Gat and Bowser (1991) equation describing the movement of water through a sequence of
pools in series. However this model would further increase the estimates of $\Delta'^{18}O_{LW-IW}$. The
predicted $^{17}O\text{-excess}_e$ displayed in Table S3 was either higher or lower than the observed $^{17}O\text{-}$
$\text{excess}_{e\ LW-IW}$. Predicted $\theta_{LW-IW}$ increased with RH from 0.521 to 0.529 which is far from the
observed values averaging 0.519. The predicted value of 0.529 at 100 % RH reflects pure
equilibrium in a situation where irrigation water and water vapor are assumed to have similar
isotope composition since irrigation water is directly vaporized into the chamber (table S3),
without any fractionation. Sensitivity tests show that regardless of the model chosen (Buhay et al.,
1996; Cernusak et al., 2016; Li et al., 2017), estimations of $\theta_{LW-IW}$ are very dependant on the isotope
compositions of the water vapor (Li et al., 2017), not measured either in our experiment or in
previous studies (Landais et al., 2006; Li et al., 2017). In the natural environment, a first order
approximation for the isotope composition of water vapor is to consider equilibrium with
precipitation. As a result of water-vapor equilibrium fractionation and soil water $^{18}O$-enrichment,
this can lead to a water vapor $^{18}O$-depleted by 10-13 ‰ compared to the soil water (Landais et al.,
2006; Lehmann et al., 2018). In this case the predicted $\lambda_{transp}$ (equivalent to $\lambda_{LW-SW}$) decreases with
increasing humidity. Finally, because wrong values of the isotope compositions of the water vapor
may affect significantly the calculation of $\Delta'^{18}O_{LW-IW}$, $\Delta'^{17}O\text{-excess}_{e\ LW-IW}$ and $\theta_{LW-SW}$, we call for
vapor isotope measurements as a prerequisite to accurately model the leaf water triple oxygen
isotope evolution with RH. However, overall, despite the uncertainties on the predicted evolution
of $\lambda_{LW-SW}$ or $\theta_{LW-SW}$ with RH, the predicted value of $^{17}O\text{-excess}_{e\ LW-IW}$ decreases when RH increases,
which is also observed, as well as reflected in the triple isotope composition of phytoliths, as
discussed below.

## 496 4.2   Imprint of changes in atmospheric RH on the $^{17}O$-excess of phytoliths

Polymerization of silica is supposed to occur in isotope equilibrium with the forming-water, and
therefore, to be only governed by temperature and the isotope composition of the forming water.
Almost a dozen temperature-dependant relationships have been empirically established between
the $\delta^{18}O$ of quartz, sinters, cherts, diatoms or phytoliths and the $\delta^{18}O$ of their forming water
($\delta^{18}O_{PhytoFW}$). Although the obtained fractionation coefficients are close (from -0.2 to -0.4 ‰ °C$^{-1}$
), the range of fractionation ($\Delta^{18}O_{Phyto-PhytoFW}$) is large (see synthesis in Alexandre et al., 2012).
The $\Delta^{18}O_{Phyto-LW}$ values obtained in the frame of the growth chamber experiment (ranging from
$27.9 \pm 7.2$ to $32.3 \pm 2.2$‰) encompass the $\Delta^{18}O_{Phyto-PhytoFW}$ of 31.1‰ calculated from the Dodd
and Sharp (2010) relationship for 25°C. It is lower than the values of 36.4 and 36 ‰ at 25 °C,
calculated from Sharp et al. (2016) and Alexandre et al. (2012). Whereas Alexandre et al. (2012)
and Sharp et al. (2016) generally estimated the forming-water $\delta^{18}O$ values, Dodd and Sharp (2010)
measured the the $\delta^{18}O$ values of the water samples. The proximity of the obtained range of
$\Delta^{18}O_{Phyto-LW}$ values to the $\Delta^{18}O_{Phyto-Phyto\ FW}$ calculated from Dodd and Sharp (2010) suggests that
phytoliths formed in equilibrium with a water of isotope composition close to that of the bulk leaf
water. This is additionally supported by the obtained averaged value of $\theta_{Phyto-LW}$ ($0.522 \pm 0.001$)
close to the $\theta_{SiO2-water}$ equilibrium value of 0.524 calculated for 25 °C from Sharp et al. (2016).
Evolution of the triple isotope composition of bulk leaf water and phytoliths can be illustrated by
plotting $\delta'^{17}O$ vs $\delta'^{18}O$, or $^{17}O$-excess vs $\delta'^{18}O$ (fig. 6) which is more appropriate to evidence small
variations. Figure 6 shows that the leaf water evolved from the irrigation water pool, becomes
increasingly subject to kinetic fractionation when RH decreased. This evolution follows a single
leaf water line reflecting $\lambda_{LW-IW} = 0.518$ or $\theta = 0.519$ (Table1). Then, if phytoliths polymerized
from the bulk leaf waters, at 25°C, according to a constant equilibrium fractionation, their expected
isotope signature should follow a line parallel to the leaf water line. This is the case for phytoliths
formed at RH higher than 40%. However, the isotope signature of phytoliths formed at 40% RH
suggest a forming water more evaporated than the bulk leaf water. The Péclet effect, which is
known to scale with transpiration (e.g. Barnard et al., 2007) can explain this discrepancy.
Advection of less evaporated stem water may decrease $\delta'^{18}O_{LW}$ and increase $^{17}O$-excess$_{LW}$ relative
to $\delta'^{18}O$ and $^{17}O$-excess of the epidermal water prone to evaporation and from which phytoliths
formed. At this point, the data scattering prevents further discussion but the possibility that when
RH is low, or when transpiration is high, the phytolith forming-water is different from the bulk
leaf water must be investigated in future research developments.
With regard to the natural samples, whereas no relationship was found between $\delta'^{18}O_{phyto}$ and RH,
a clear positive linear dependency of $^{17}O$-excess$_{phyto}$ to RH was shown, equivalent to 2.1 per meg
/ % when the annual RH average was taken into account, or to 3.4 per meg / % when the average
of the growing season (RH-rd0>1) was taken into account (fig. 4). These coefficients are close to
the slope of the lines obtained for the growth chamber experiment between $^{17}O$-excess$_{Phyto}$, $^{17}O$-
excess$_{e\ LW-IW}$ and $^{17}O$-excess$_{e\ Phyto-IW}$ and 80 to 40% RH (fig. 1a, e and f). This consistency
represents a major positive step in examining whether changes in atmospheric RH imprint the $^{17}O$-
excess of natural phytolith assemblages in a predictable way. Without taking into account the two
outliers, the linear regression between RH-rd0>1 and $^{17}O$-excess$_{phyto}$ for a 95% confidence interval
can be expressed as follows:
RH-rd0>1 = 0.14 ± 0.02 (S.E) x $^{17}$O-excess$_{phyto}$ + 100.5 ± 4.7 (S.E)            Eq. 2
where $^{17}$O-excess$_{phyto}$ is expressed in per meg and RH in %, $r^2$ = 0.48, and p < 0.001. S.E. stands
for standard error. The S.E. of the predicted RH-rd0>1value is ± 5.6%. However, the data
scattering (fig. 4) call for assessing additional parameters that can contribute to changes in $^{17}$O-
excess$_{Phyto}$, beside RH, before using the $^{17}$O-excess$_{phyto}$ for quantitative RH reconstruction.
One can expect that the isotope composition of the soil water taken-up by the roots impacts $^{17}$O-
excess$_{Phyto}$. In tropical dry and humid areas, evaporative kinetic fractionation can lead to a $^{18}$O-
enrichment of the soil water of several ‰, in the first dm depth (e.g. Gaj et al., 2016; Liu et al.,
2010). Spatial variability in the composition of the rainfall feeding the upper soil water may also
intervene. However, the amount-weighted values of δ'$^{18}$O$_{Pre}$ along the sampled transect vary little
(Table 2). With regard to $^{17}$O-excess, changes in soil water evaporation rather than the small
variations expected for $^{17}$O-excess$_{Pre}$ (Landais et al., 2010b; Li et al., 2015) should impact the
evolution of $^{17}$O-excess$_{Phyto}$, although, here, the lack of measurements only allow for speculation.
The vegetation type and the plant part from which phytoliths come from may also bring some
noise to the relationship between $^{17}$O-excess$_{phyto}$ and RH. In grasses,  leaf water is expected to be
more prone to evaporative enrichment than stem water, and inside the leaf itself, the heterogeneity
of evaporative sites repartition and water movements can lead to a significant heterogeneity in the
δ$^{18}$O signatures of water and phytoliths (Cernusak et al., 2016; Helliker and Ehleringer, 2000;
Webb and Longstaffe, 2002). However, soil top phytolith assemblages likely record several
decades of annual bulk phytolith production and their isotope composition is expected to be an
average. This would explain the consistency of the $^{17}$O-excess$_{Phyto}$ data obtained from bulk grass
phytoliths from climate chambers and the bulk phytolith assemblages from natural vegetation.
Further investigation on the extent of the heterogeneity of $^{17}$O-excess in water and phytoliths in
mature grasses would help to verify this assumption. In trees, the Globular granulate phytolith is
assumed to come from the non-transpiring secondary xylem of the wood. Thus Globular granulate
phytoliths should present an isotope signature closer to that of the soil water, or less impacted by
kinetic fractionation than grass phytoliths. However, for a given range of RH, samples with
significant representations of both phytolith categories (i.e wooded savanna and enclosed savanna
samples with d/p from 0.1 to 1.6) present $^{17}$O-excess values close to the values obtained by samples
with very low or very high d/p (figs. 4 and 5). To further assess the significance of the Globular
granulate isotope signature, we calculated δ'$^{18}$O$_{PhytoFW}$ values (Table 2) using the Dodd and Sharp
(2010) fractionation factor and compared it to the precipitation-weighted δ'$^{18}$O$_{Pre-rd0>1}$ average.  For
the humid forest assemblages, δ'$^{18}$O$_{PhytoFW}$ values are higher than δ'$^{18}$O$_{pre\ rd0>1}$ by 4.6 ± 1.5 ‰.  This
difference is larger than the range of $^{18}$O-enrichment observed for the upper 10 cm depth of soil
water under tropical humid forests (2-3‰; Liu et al., 2008; Stahl et al., 2013), suggesting that
evaporative isotope signatures of both soils and leaf water imprinted the Globular granulate
phytolith type. This is in  line with recent $^{18}$O-labelling experiment showing that the $^{18}$O-enriched
oak phloem water may exchange with xylem water under low transpiration rates (Lehmann et al.,
2018). Complementary examination of the isotope signature of phytolith assemblages from forests
growing under different RH conditions (i.e dry forests, humid forests, rainforests), as well as
further investigation of the anatomical origin of the Globular granulate phytolith type are now
required to further discuss the meaning of the $^{17}$O-excess signal brought by wooded savanna and
tropical forest phytolith assemblages.
Biases due to the calibration methodology may also be responsible for the data scattering.
Imperfect adequacy between the space scales recorded by the soil top phytolith assemblages and
the RH variables may come into play. Phytolith assemblages represent a mixture of local and wind-
transported phytoliths. In the open saharian, sahelian and soudanian zones of West Africa the
winter low altitude north-easterly trade winds may transport phytoliths southward, reducing
differences between assemblages from different biogeographic zones and increasing differences
among assemblages of a given biogeographic zone (Bremond et al., 2005b). Additional samples
from other geographic zones are thus needed to increase the robustness of the relationship. With
regard to the recorded time scales, the CRU RH 30 years averages are in agreement with the several
decades of phytolith production.

## 591 5 Conclusion

The present combination of growth chamber and *in situ* transect calibrations lay the groundwork
for further examination of the robustness of the $^{17}$O-excess$_{Phyto}$ as a proxy of changes in RH. The
growth chamber experiment demonstrated that change in RH imprints $^{17}$O-excess$_{Phyto}$ (by 4.1 per
meg / % between 40 and 80% RH) or the $^{17}$O-excess$_{e\ Phyto-IW}$ (by 4.3 per meg / %, between 40 and
80% RH) through its imprint on $^{17}$O-excess$_{e\ LW-IW}$. As the isotope composition of the irrigation
water was stable, and transpiration likely reached a steady state, the positive correlation between
$^{17}$O-excess$_{LW}$ and RH was only governed by the kinetic fractionation occurring in the leaf
epidermis water subject to evaporation, as supported by the value of $\theta_{LW-IW}$ of 0.517, close to $\theta_{diff}$.
In order to model the triple oxygen isotope fractionation in play at the soil/plant/atmosphere
interface we require direct and continuous measurements of the triple isotope composition of water
vapor. Such measurements should develop in the near futur through the use of isotope ratio infrared
analyzers (e.g. Berkelhammer et al., 2013; Schmidt et al., 2010). We also suggest to constrain as
much as possible the isotope composition of the soil water taken up by the roots. Stem water is
usually used as an analogue of soil water when modelling $\delta'^{17}O_{LW}$ and $\delta'^{18}O_{LW}$ (Landais et al.,
2006; Li et al., 2017). However, in the stem, water in the phloem that is bidirectional (moves up
and down the plant's stem) receives the contribution of evaporating leaf water, and water in the
xylem that is unidirectional (moves up the plant's stem) may exchange with phloem waters
(Lehmann et al., 2018). Consequently one may expect the isotope composition of stem water to
be slightly different than that of soil water (Berkelhammer et al., 2013; Treydte et al., 2014).
When plotting $^{17}$O-excess$_{Phyto}$ *vs* RH, the samples collected along the West and Central African
relative humidity transect define a correlation coefficient ranging from 2.1 to 3.4 per meg / %
(depending on the RH variable taken into account) and lay close to the growth chamber $^{17}$O-
excess$_{Phyto}$ line. This supports that RH is an important control of $^{17}$O-excess$_{Phyto}$ in natural
environment, even if phytolith assemblages come from different vegetation types. However, other
parameters such as changes in the triple isotope composition of the soil water, vegetation source
or imperfect adequation between the space scales recorded by the soil top phytolith assemblages
and the RH variables may come into play and explain the scattering of $^{17}O$-excess$_{Phyto}$. Assessment
of these parameters through additional growth chambers experiments and field campaigns will
bring us closer to an accurate proxy of changes in relative humidity.

*Acknowledgements*
This study was supported by the French program INSU-LEFE and benefited from the CNRS
human and technical resources allocated to the ECOTRONS Research Infrastructures as well as
from the state allocation 'Investissements d'Avenir' ANR-11-INBS-0001.

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

to accompany the Unesco/AETFAT/UNSO vegetation map of Africa (Unesco).

Table 1. Growth chamber experiment : experimental set-up, phytolith content and morphological characteristics, isotope enrichments ($\Delta'^{*}_{A-B} = {}^{*}\delta'_a - {}^{*}\delta'_b$), associated $^{17}$O-excess$_e$ ($^{17}$O-excess$_e = \Delta'^{17} - 0.528 \times \Delta'^{18}$), θ (θ = $\Delta'^{17} / \Delta'^{18}$) and λ values of phytoliths compared to either leaf water or irrigation water and of leaf water compared to irrigation water. Av : average ; n : number of replicates ;  SD : standard deviation calculated on the replicates; n.v. : no value. Transp. (l/day), Conc. (% d.w.) and LC (%) stands for transpiration expressed in liter/day, phytolith concentration expressed in % of the dry weight and long cell abundance in the phytolith morphological assemblage expressed in % of counted phytoliths with taxonomic significance, respectively. Samples are named according to the climate chamber # they were collected in (e.g. P1, P2), the set relative humidity (e.g. 40, 60) and the date of sampling (e.g. 29-04-16 for dd/mm/yy).

| Experimental set-up | | | | | | | | Phytoliths (Phyto) | | | Leaf water - irrigation water (LW-IW) | | | | Phytolith - leaf water (Phyto-LW) | | | | Phytolith - irrigation water (Phyto-IW) | | | |
|---|---|---|---|---|---|---|---|---|---|---|---|---|---|---|---|---|---|---|---|---|---|---|
| Duration | Temp. | SD | RH | SD | Light | Transp. | Biomass | Sample | Conc. | LC | $\Delta'^{18}$O | $\Delta'^{17}$O | $^{17}$O-excess$_e$ | θ | $\Delta'^{18}$O | $\Delta'^{17}$O | $^{17}$O-excess$_e$ | θ | $\Delta'^{18}$O | $\Delta'^{17}$O | $^{17}$O-excess$_e$ | θ |
| day | °C | | % | | mmol/m²/sec | l/day | g | | % d.w. | % | ‰ | ‰ | per meg | | ‰ | ‰ | per meg | | ‰ | ‰ | per meg | |
| 11 | 25 | 0.2 | 41.2 | 1 | 278 | | 13 | P1-40-29-04-16 | n.v. | | 16.238 | 8.420 | -154 | 0.519 | 33.776 | 17.589 | -244 | 0.521 | 50.013 | 26.009 | -398 | 0.520 |
| 10 | 25 | 0.2 | 41.3 | 1.1 | 278 | 0.49 | 21 | P10-40-10-05-16 | 0.8 | | 13.171 | 6.799 | -155 | 0.516 | 33.530 | 17.498 | -206 | 0.522 | 46.701 | 24.297 | -361 | 0.520 |
| 11 | 25 | 0.4 | 41.9 | 1 | 311 | 0.69 | 37 | P1-40-20-05-16 | 0.8 | 21 | 16.345 | 8.460 | -170 | 0.518 | 29.577 | 15.401 | -216 | 0.521 | 45.922 | 23.861 | -385 | 0.520 |
| 14 | 25 | 0.2 | 41.4 | 0.9 | 278 | 0.65 | 38 | P1-40-03-06-16 | 1.8 | | n.v. | n.v. | n.v. | n.v. | 32.415 | 16.874 | -241 | 0.521 | n.v. | n.v. | n.v. | n.v. |
| | | | | | | Av. | 0.61 | | 1.2 | | 15.251 | 7.893 | -159 | 0.517 | 32.324 | 16.840 | -227 | 0.521 | 47.545 | 24.723 | -381 | 0.520 |
| | | | | | | SD | 0.11 | | 0.6 | | 1.802 | 0.947 | 9 | 0.001 | 1.925 | 1.011 | 19 | 0.0006 | 2.172 | 1.135 | 19 | 0.0003 |
| 11 | 25 | 0.5 | 60.2 | 2.5 | 311 | | 21 | P10-60-29-04-16 | n.v. | | 15.115 | 7.864 | -117 | 0.520 | 29.133 | 15.211 | -171 | 0.522 | 44.248 | 23.075 | -288 | 0.521 |
| 11 | 25 | 0.2 | 60.5 | 1 | 289 | 0.57 | 33 | P2-60-10-05-16 | 0.7 | | 16.885 | 8.737 | -178 | 0.517 | 25.877 | 13.575 | -88 | 0.525 | 42.761 | 22.312 | -266 | 0.522 |
| 10 | 25 | 0.8 | 60.2 | 4.8 | 311 | 0.60 | 48 | P10-60-20-05-16 | 0.8 | 13 | 12.014 | 6.242 | -101 | 0.520 | 30.254 | 15.804 | -170 | 0.522 | 42.268 | 22.047 | -271 | 0.522 |
| 14 | 25 | 0.6 | 60.3 | 3.2 | 311 | 0.76 | 60 | P10-60-03-06-16 | 1.3 | | n.v. | n.v. | n.v. | n.v. | 32.915 | 17.186 | -193 | 0.522 | n.v. | n.v. | n.v. | n.v. |
| | | | | | | Av. | 0.64 | | 0.9 | | 14.671 | 7.614 | -132 | 0.519 | 29.545 | 15.444 | -156 | 0.523 | 43.093 | 22.478 | -275 | 0.522 |
| | | | | | | SD | 0.10 | | 0.3 | | 2.465 | 1.266 | 41 | 0.001 | 2.915 | 1.496 | 46 | 0.0012 | 1.031 | 0.534 | 11 | 0.0001 |
| 11 | 25 | 0.2 | 80.2 | 2.8 | 289 | | 24 | P2-85-29-04-16 | n.v. | | 7.826 | 4.067 | -65 | 0.520 | 28.039 | 14.668 | -136 | 0.523 | 35.865 | 18.736 | -201 | 0.522 |
| 10 | 25 | 0.2 | 81.5 | 1.3 | 289 | 0.28 | 27 | P1-85-10-05-16 | 0.4 | | 7.957 | 4.139 | -62 | 0.520 | 28.276 | 14.783 | -147 | 0.523 | 36.233 | 18.922 | -209 | 0.522 |
| 11 | 25 | 0.2 | 76.6 | 2.5 | 278 | 0.22 | 27 | P2-85-20-05-16 | 0.6 | 10 | 6.679 | 3.429 | -97 | 0.513 | 28.668 | 14.993 | -144 | 0.523 | 35.347 | 18.422 | -241 | 0.521 |
| 14 | 25 | 0.2 | 82.5 | 1.1 | 289 | 0.36 | 37 | P2-85-03-06-16 | 1.0 | | n.v. | n.v. | n.v. | n.v. | 28.888 | 15.041 | -212 | 0.521 | n.v. | n.v. | n.v. | n.v. |
| | | | | | | Av. | 0.29 | | 0.7 | | 7.487 | 3.879 | -75 | 0.518 | 28.468 | 14.871 | -160 | 0.522 | 35.815 | 18.694 | -217 | 0.522 |
| | | | | | | SD | 0.07 | | 0.3 | | 0.703 | 0.391 | 20 | 0.004 | 0.382 | 0.176 | 35 | 0.0012 | 0.445 | 0.253 | 21 | 0.0007 |
| 11 | 25 | | 100.0 | | 307 | 0.03 | 31 | P3-100-10-05-16 | 0.0 | | 14.681 | 7.630 | -122 | 0.520 | 21.325 | 11.170 | -90 | 0.524 | 36.006 | 18.800 | -212 | 0.522 |
| 10 | 25 | | 100.0 | | 307 | 0.01 | | P3-100-20-05-16 | 0.0 | 5 | 7.706 | 4.014 | -54 | 0.521 | 27.344 | 14.284 | -153 | 0.522 | 35.050 | 18.299 | -208 | 0.522 |
| 14 | 25 | | 100.0 | | 307 | 0.05 | 21 | P3-100-03-06-16 | 0.2 | | n.v. | n.v. | n.v. | n.v. | 35.233 | 18.403 | -200 | 0.522 | n.v. | n.v. | n.v. | n.v. |
| | | | | | | Av. | 0.03 | | 0.1 | | 11.194 | 5.822 | -88 | 0.520 | 27.968 | 14.619 | -148 | 0.523 | 35.528 | 18.549 | -210 | 0.522 |
| | | | | | | SD | 0.02 | | 0.1 | | 4.932 | 2.557 | 48 | 0.001 | 6.975 | 3.628 | 55 | 0.0008 | 0.676 | 0.354 | 3 | 0.0000 |
| | | | | | | | | Av.(a) | | | | | | 0.519 | | | | 0.522 | | | | |
| | | | | | | | | SD (a) | | | | | | 0.002 | | | | 0.001 | | | | |
| | | | | | | | | | | | | | λ=0.518 | | | | | | | | λ=0.515 | |


**Table 2.** Natural West and Central African phytolith samples: coordinates, climatic parameters,
calculated phytolith index d/p, measured $\delta'^{18}O_{Phyto}$, $\delta'^{17}O_{Phyto}$, $^{17}O$-excess$_{Phyto}$, calculated $\delta'^{18}O$
of phytolith forming water ($\delta^{18}O_{PhytoFW}$) and precipitation-weighted $\delta'^{18}O_{Pre-rd0>1}$. Average and
standard deviation (SD) are given for replicates. MAP: Mean Annual Precipitation; MAT:
Mean Annual Temperature; RH: mean annual relative humidity; RH15: RH at 15:00 H UTC;
RH-rd0>1: relative humidity average for months with at least one day with precipitation higher
than 0.1mm; RH15-rd0>1: RHrd0>1 at 15:00 H UTC. See text for data source and calculation.

| Identifier | Lat | long | MAP | MAT | RH | RH-rd0>1 | RH15 | RH15-rd0>1 | $\delta'^{18}O_{Pre}$ (1) | d/p | n | $\delta'^{18}O_{Phyto}$ | SD | $\delta'^{17}O_{Phyto}$ | SD | $^{17}O$-excess$_{Phyto}$ | SD | $\delta'^{18}O_{Phyto\ FW}$ | $\Delta'^{18}O_{Phyto-Pre-rd0>1}$ |
|---|---|---|---|---|---|---|---|---|---|---|---|---|---|---|---|---|---|---|---|
| | | | mm | °C | % | % | % | % | ‰ | | | ‰ | | ‰ | | per meg | | ‰ | ‰ |
| **Savana** | | | | | | | | | | | | | | | | | | | |
| RIM 3 | 21.5 | -13.0 | 52.4 | 27.3 | 47.1 | 61.7 | 35.4 | 47.0 | -3.220 | 0.03 | | 33.127 | | 17.218 | | -243 | | 2.384 | 36.351 |
| RIM 8 | 21.0 | -12.2 | 49.1 | 28.2 | 44.1 | 60.5 | 33.0 | 45.9 | -3.420 | 0.04 | | 34.813 | | 18.304 | | -243 | | 4.221 | 38.239 |
| MAU06 | 20.6 | -12.6 | 68.8 | 27.6 | 44.0 | 58.0 | 33.0 | 44.1 | -3.829 | 0.04 | | 28.871 | | 15.088 | | -268 | | -1.816 | 32.707 |
| RIM 11 | 16.9 | -15.2 | 209.1 | 27.3 | 45.9 | 68.5 | 32.5 | 52.2 | -4.047 | 0.04 | | 37.506 | | 19.785 | | -211 | | 6.745 | 41.561 |
| RIM 10 | 16.7 | -15.2 | 227.6 | 27.2 | 45.7 | 68.7 | 32.1 | 52.1 | -4.042 | 0.01 | | 38.163 | | 20.094 | | -256 | | 7.377 | 42.214 |
| S33 | 16.4 | -14.8 | 270.5 | 27.7 | 42.7 | 57.6 | 29.7 | 41.8 | -3.861 | 0.04 | | 35.961 | | 18.939 | | -225 | | 5.276 | 39.829 |
| S32 | 16.3 | -15.4 | 284.4 | 27.3 | 46.9 | 61.6 | 33.5 | 46.2 | -3.768 | 0.04 | | 37.297 | | 19.617 | | -266 | | 6.537 | 41.072 |
| C4L1 | 16.1 | -14.0 | 287.7 | 29.8 | 40.9 | 57.1 | 29.4 | 42.9 | -3.874 | 0.02 | 2 | 34.915 | 0.368 | 18.340 | 0.203 | -262 | 11 | 4.609 | 38.797 |
| S40 | 16.1 | -13.9 | 329.1 | 29.2 | 40.6 | 56.8 | 29.4 | 43.0 | -3.969 | 0.06 | | 35.385 | | 18.592 | | -262 | | 4.967 | 39.363 |
| S29 | 16.1 | -14.9 | 313.0 | 27.8 | 43.6 | 59.1 | 30.8 | 43.7 | -3.833 | 0.05 | 2 | 35.449 | 0.583 | 18.653 | 0.303 | -236 | 0 | 4.785 | 39.290 |
| 82-46 | 16.0 | -16.0 | 316.4 | 27.1 | 53.0 | 67.5 | 40.1 | 54.2 | -3.604 | 0.03 | | 33.575 | | 17.654 | | -228 | | 2.800 | 37.185 |
| 82-47 | 16.0 | -16.0 | 316.4 | 27.1 | 53.0 | 67.5 | 40.1 | 54.2 | -3.604 | 0.04 | | 36.429 | | 19.169 | | -247 | | 5.642 | 40.039 |
| S44 | 15.8 | -13.5 | 369.1 | 29.6 | 40.2 | 57.2 | 29.6 | 44.1 | -4.073 | 0.04 | 2 | 36.211 | 0.593 | 19.041 | 0.284 | -258 | 24 | 5.863 | 40.292 |
| C4L3 | 15.4 | -13.7 | 467.7 | 29.6 | 41.2 | 59.1 | 30.3 | 45.7 | -4.023 | 0.05 | 2 | 33.688 | 0.312 | 17.652 | 0.175 | -290 | 13 | 3.345 | 37.719 |
| S54 | 15.3 | -13.0 | 443.6 | 29.7 | 41.3 | 60.0 | 31.0 | 47.2 | -4.009 | 0.04 | | 35.586 | | 18.680 | | -282 | | 5.261 | 39.603 |
| S58 | 15.1 | -12.8 | 478.6 | 29.7 | 42.0 | 56.3 | 31.7 | 44.3 | -4.009 | 0.05 | 2 | 36.161 | 0.234 | 19.006 | 0.143 | -266 | 21 | 5.833 | 40.179 |
| C5L1 | 15.0 | -12.9 | 583.2 | 29.7 | 42.5 | 57.2 | 32.1 | 44.9 | -3.972 | 0.06 | 3 | 29.525 | 0.483 | 15.500 | 0.257 | -208 | 7 | -0.787 | 33.505 |
| 83-62 | 14.9 | -12.3 | 515.8 | 29.7 | 42.9 | 58.1 | 32.6 | 46.0 | -4.097 | 0.06 | 2 | 36.320 | 0.747 | 19.095 | 0.424 | -262 | 36 | 5.987 | 40.426 |
| S5 | 14.7 | -16.2 | 511.1 | 28.1 | 53.3 | 68.6 | 39.2 | 53.9 | -3.789 | 0.08 | 2 | 24.297 | 0.115 | 12.704 | 0.064 | -205 | 4 | -6.312 | 28.094 |
| 82-79 | 14.2 | -16.1 | 669.0 | 28.3 | 54.2 | 70.1 | 39.7 | 55.2 | -3.774 | 0.03 | 2 | 33.913 | 0.046 | 17.798 | 0.076 | -229 | | 3.356 | 37.694 |
| 83-75 | 14.1 | -12.7 | 736.2 | 29.1 | 46.7 | 63.1 | 35.6 | 50.3 | -3.936 | 0.03 | | 32.418 | | 16.969 | | -290 | | 2.000 | 36.362 |
| 82-78 | 14.1 | -16.1 | 669.0 | 28.3 | 55.2 | 71.0 | 40.8 | 56.2 | -3.768 | 0.18 | | 23.789 | | 12.437 | | -201 | | -6.785 | 27.565 |
| S84 | 13.9 | -13.4 | 775.2 | 28.9 | 47.4 | 64.1 | 36.1 | 50.9 | -4.040 | 0.03 | 2 | 32.600 | 0.435 | 17.080 | 0.221 | -277 | 5 | 2.141 | 36.648 |
| S118 | 13.6 | -13.7 | 878.1 | 28.6 | 49.6 | 66.3 | 37.7 | 52.9 | -4.008 | 0.02 | | 30.007 | | 15.779 | | -188 | | -0.501 | 34.023 |
| S88 | 13.6 | -13.6 | 880.0 | 28.6 | 49.4 | 66.2 | 37.7 | 52.9 | -3.996 | 0.02 | | 28.371 | | 14.900 | | -189 | | -2.129 | 32.375 |
| 83-120 | 13.5 | -13.8 | 934.5 | 28.5 | 50.7 | 67.3 | 38.7 | 53.8 | -3.984 | 0.05 | | 31.622 | | 16.570 | | -262 | | 1.101 | 35.614 |
| 83-122 | 13.4 | -14.9 | 947.3 | 28.1 | 53.6 | 68.1 | 39.8 | 53.7 | -3.928 | 0.03 | 2 | 31.240 | 0.628 | 16.396 | 0.335 | -231 | 9 | 0.649 | 35.176 |
| S122 | 13.3 | -13.9 | 934.5 | 28.5 | 52.1 | 68.1 | 39.7 | 53.7 | -3.971 | 0.03 | | 34.379 | | 18.095 | | -219 | | 3.851 | 38.350 |
| S93 | 13.3 | -13.2 | 1005.3 | 28.6 | 51.6 | 68.2 | 39.7 | 55.1 | -3.925 | 0.08 | | 30.064 | | 15.787 | | -211 | | -0.435 | 33.989 |
| 83-98 | 13.1 | -12.8 | 1067.0 | 28.7 | 52.7 | 69.3 | 40.8 | 56.3 | -4.060 | 0.04 | | 29.692 | | 15.621 | | -177 | | -0.800 | 33.753 |
| S128 | 13.0 | -14.1 | 1055.1 | 28.2 | 54.7 | 70.2 | 41.7 | 56.5 | -3.765 | 0.07 | 2 | 34.078 | 0.567 | 17.919 | 0.265 | -233 | 29 | 3.500 | 37.843 |
| S.130 | 12.9 | -14.2 | 1113.9 | 28.0 | 55.1 | 70.3 | 41.8 | 56.4 | -3.961 | 0.03 | 3 | 35.909 | 0.515 | 18.692 | 0.254 | -268 | 19 | 5.286 | 39.870 |
| 83-103 | 12.9 | -12.4 | 1114.0 | 28.5 | 53.7 | 70.4 | 41.8 | 56.6 | -4.329 | 0.03 | | 30.499 | | 15.855 | | -249 | | -0.024 | 34.828 |
| S.138 | 12.9 | -14.9 | 1127.1 | 27.7 | 56.8 | 70.9 | 42.4 | 56.6 | -4.069 | 0.03 | 2 | 35.822 | 0.5 | 18.667 | 0.3 | -247 | 5 | 5.138 | 39.891 |
| S136 | 12.8 | -14.7 | 1113.4 | 27.8 | 57.4 | 71.5 | 43.3 | 57.4 | -4.023 | 0.02 | | 33.422 | | 17.355 | | -246 | | 2.767 | 37.445 |
| 83-116 | 12.7 | -12.2 | 1233.2 | 28.5 | 54.8 | 71.4 | 42.7 | 58.5 | -4.316 | 0.03 | 2 | 31.084 | | 16.149 | | -264 | | 0.558 | 35.401 |
| 83-115 | 12.4 | -12.3 | 1301.3 | 27.8 | 56.5 | 66.8 | 44.2 | 53.8 | -4.170 | 0.01 | 2 | 31.524 | 0.4 | 16.418 | 0.2 | -226 | 17 | 0.887 | 35.694 |
| **Wooded savana** | | | | | | | | | | | | | | | | | | | |
| 83-8 | 14.9 | -15.9 | 485.2 | 28.0 | 50.9 | 67.0 | 37.3 | 52.4 | -3.948 | 0.16 | | 36.813 | | 19.167 | | -270 | | 6.181 | 40.762 |
| S7 | 14.8 | -16.0 | 513.6 | 28.2 | 52.0 | 67.9 | 38.0 | 53.1 | -4.263 | 0.26 | | 29.491 | | 15.331 | | -241 | | -1.076 | 33.755 |
| 83-4 | 14.7 | -16.5 | 539.7 | 27.0 | 57.4 | 70.9 | 43.7 | 57.3 | -3.821 | 0.43 | | 26.127 | | 13.565 | | -231 | | -4.665 | 29.948 |
| 82-77 | 14.6 | -16.3 | 535.5 | 28.0 | 53.5 | 69.0 | 39.2 | 54.0 | -3.798 | 0.14 | 2 | 35.214 | 0.8 | 18.312 | 0.4 | -281 | 10 | 4.601 | 39.012 |
| S91 | 13.6 | -13.4 | 883.1 | 28.7 | 49.4 | 66.2 | 37.8 | 53.0 | -3.984 | 0.40 | 2 | 34.512 | 0.2 | 18.009 | 0.1 | -213 | 30 | 4.013 | 38.496 |
| C4L8 | 13.5 | -13.7 | 878.1 | 28.6 | 50.7 | 67.3 | 38.7 | 53.8 | -3.984 | 0.13 | | 32.302 | | 16.850 | | -206 | | 1.793 | 36.286 |
| 83-127 | 13.1 | -14.1 | 1055.1 | 28.2 | 53.6 | 69.3 | 40.7 | 55.5 | -3.785 | 0.49 | 2 | 35.638 | 0.2 | 18.573 | 0.1 | -244 | 1 | 5.054 | 39.423 |
| **Enclosed savana** | | | | | | | | | | | | | | | | | | | |
| Biendi 1 | -2.0 | 11.1 | 1839.0 | 25.9 | 80.9 | 80.9 | 67.4 | 67.4 | -3.687 | 0.76 | | 33.086 | 0.0 | 17.233 | 0.0 | -205 | 20 | 2.096 | 36.773 |
| Doubou | -1.8 | 10.9 | 1986.0 | 25.9 | 81.2 | 81.2 | 67.9 | 67.9 | -3.631 | 1.58 | | 31.931 | 0.6 | 16.665 | 0.3 | -194 | 8 | 0.954 | 35.562 |
| **Humid forest** | | | | | | | | | | | | | | | | | | | |
| 83-151 | 12.5 | -16.6 | 1428.6 | 26.5 | 65.8 | 74.9 | 51.0 | 60.8 | -3.787 | 6.78 | | 33.097 | | 17.288 | | -187 | | 2.221 | 36.884 |
| S155 | 12.5 | -16.3 | 1352.6 | 27.0 | 64.4 | 74.3 | 49.4 | 59.9 | -3.777 | 4.85 | 2 | 29.092 | 0.2 | 15.180 | 0.1 | -181 | 4 | -1.698 | 32.869 |
| 04-94 | | 13.1 | 1676.4 | 24.3 | 81.4 | 81.4 | 65.8 | 65.8 | -4.464 | 2.44 | 2 | 32.638 | 0.2 | 17.093 | 0.1 | -140 | 4 | 1.371 | 37.102 |
| 04-88 | | 12.4 | 1707.0 | 24.9 | 81.9 | 81.9 | 67.1 | 67.1 | -4.458 | 5.45 | | 33.137 | 0.4 | 17.345 | 0.2 | -151 | 18 | 1.977 | 37.595 |
| 04-47 | -0.2 | 12.3 | 1724.0 | 26.2 | 82.1 | 82.1 | 67.4 | 67.4 | -1.515 | 3.48 | | 32.953 | 0.0 | 17.215 | 0.0 | -185 | 17 | 2.026 | 34.468 |
| 04-66 | -0.2 | 12.5 | 1690.6 | 25.8 | 82.0 | 82.0 | 67.3 | 67.3 | -4.354 | 1.84 | | 29.959 | 0.6 | 15.641 | 0.3 | -177 | 24 | -1.040 | 34.314 |
| 04-65 | -0.2 | 12.6 | 1690.6 | 25.8 | 82.0 | 82.0 | 67.3 | 67.3 | -4.195 | 2.19 | | 32.791 | 0.6 | 17.158 | 0.3 | -156 | 25 | 1.791 | 36.985 |
| 04-118 | -0.2 | 10.5 | 2148.4 | 26.4 | 82.5 | 82.5 | 69.2 | 69.2 | -3.556 | 3.69 | | 31.840 | 0.4 | 16.648 | 0.2 | -164 | 4 | 0.945 | 35.396 |
| Dimoniki | -4.1 | 12.4 | 1286.6 | 24.7 | 80.3 | 80.3 | 68.1 | 68.1 | -4.284 | 5.80 | | 30.928 | | 16.123 | | -205 | | -0.275 | 35.212 |
| **Outliers** | | | | | | | | | | | | | | | | | | | |
| RIM1 | 16.7 | -16.0 | 216.4 | 26.7 | 52.3 | 72.2 | 39.4 | 58.6 | -3.857 | 0.06 | | 38.131 | | 19.828 | | -305 | | 7.264 | 41.987 |
| C3L4 | 15.6 | -14.2 | 362.0 | 29.3 | 41.8 | 59.1 | 30.3 | 45.0 | -3.968 | 0.06 | | 25.185 | 0.2 | 13.141 | 0.1 | -157 | 10 | -5.211 | 29.153 |

(1) Amount weighted average for months with at least one day with precipitation>0.1mm

**Figure captions**


**Figure 1.** Growth chamber experiment: a) $^{17}$O-excess *vs* relative humidity (RH) of irrigation
water (IW), soil water (SW), leaf water (LW) and phytolith (Phyto). Error bars show standard
deviation (SD) on the replicates. They are smaller than the symbol when not shown. b) $^{18}$O-
enrichment from irrigation water to leaf water ($\Delta'^{18}O_{LW-IW}$), from irrigation water to phytolith
($\Delta'^{18}O_{Phyto-IW}$) and from leaf water to phytolith ($\Delta'^{18}O_{Phyto-LW}$). c) $^{17}$O-excess associated with the
enrichment from irrigation water to leaf water ($^{17}O-excess_{e\ LW-IW}$), from irrigation water to
phytolith ($^{17}O-excess_{e\ Phyto-IW}$), and from leaf water to phytolith ($^{17}O-excess_{e\ Phyto-LW}$). d, e and
f) linear correlations for the 40-80% RH range extracted from a, b and c, respectively.
**Figure 2.** Growth chamber experiment: phytolith types extracted from *Festuca arundinaceae*
and observed in natural light microscopy: epidermal long cell (LC), epidermal short cell (SC).
**Figure 3.** Natural West and Central African transect: $\delta'^{18}O$ of phytoliths ($\delta'^{18}O_{Phyto}$) *vs* relative
humidity RH-rd0>1 (see fig. 4 for explanation). Error bars show standard deviation (SD) on the
replicates. When not shown, they are smaller than the symbol.
**Figure 4.** Natural West and Central African transect: $^{17}$O-excess *vs* relative humidity (RH) of
phytolith assemblages from soil tops collected under savanna, wooded savanna, humid forest
and enclosed savanna along a humidity gradient (Table 1). The growth chamber $^{17}O-excess_{Phyto}$
*vs* RH correlation line is displayed for comparison. a) RH-Av: yearly average of monthly
means; b) RH-rd0>1: yearly average of monthly means for months with at least one day with
precipitation higher than 0.1mm; c) RH15: RH at 15:00 H UTC; d) RH15-rd0>1: RH-rd0>1 at
15:00 H UTC.
**Figure 5.** Natural West and Central African transect: $^{17}$O-excess of phytoliths ($^{17}O-excess_{Phyto}$)
*vs* d/p.
**Figure 6.** Growth chamber experiment: $^{17}$O-excess *vs* $\delta'^{18}O$ of irrigation water (IW), soil water
(SW), bulk leaf water (LW) and phytolith (Phyto). Error bars show standard deviation (SD) on
the replicates. The leaf water line (blue) represents how the triple oxygen isotope composition
of the bulk leaf water of *Festuca arundinacea* evolves from an irrigation water signature to a
more evaporated water signature when RH decreases. This evolution follows a slope equivalent
to $\theta=0.518$ in a $\Delta'^{17}O$ vs $\Delta'^{18}O$ space (table 1). Assuming that phytoliths precipitate from the
bulk leaf water, the expected phytolith line (black) should be parallel to the leaf water line as
the equilibrium fractionation between phytolith and leaf water is constant at constant
temperature (25°C). In the investigated case this fractionation, represented by the black dotted
line, is equivalent to $\theta=0.522$ (table 1). The isotope signature of phytoliths formed at RH higher
than 40% follow the expected phytolith line. However, the isotope signature of phytoliths
formed at 40% RH suggest a forming water more evaporated than the bulk leaf water.

**Figure 1**

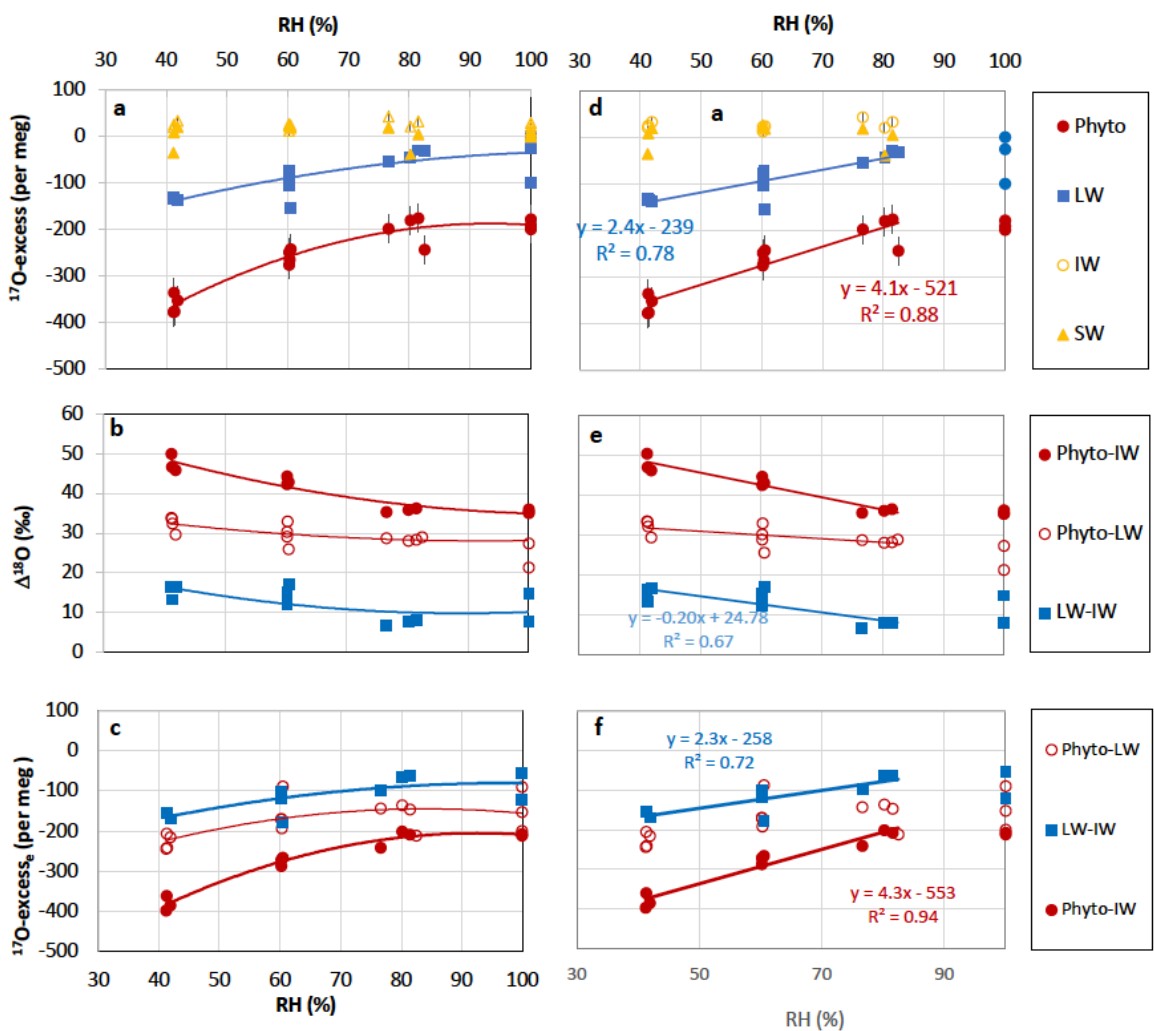


**Figure 2**

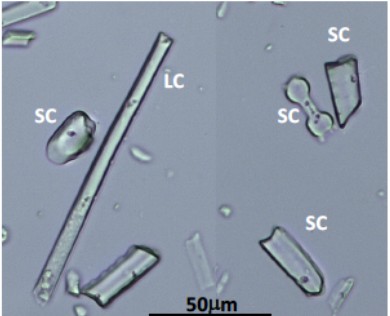



**Figure 3**

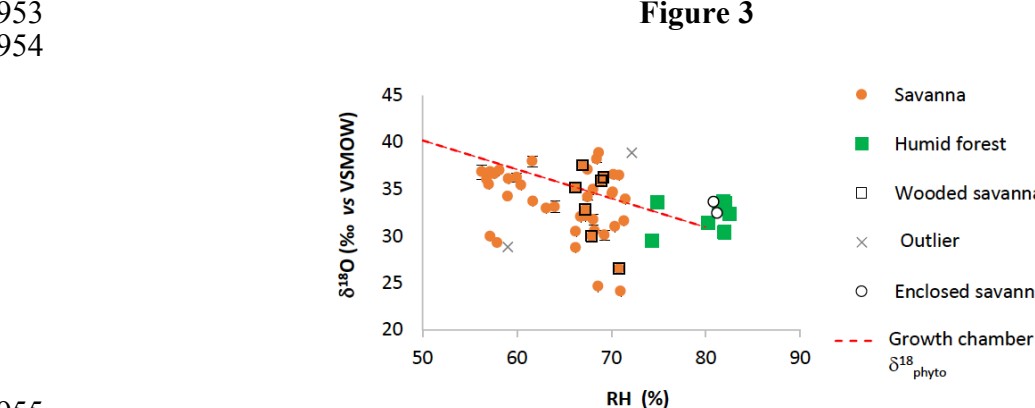


**Figure 4**

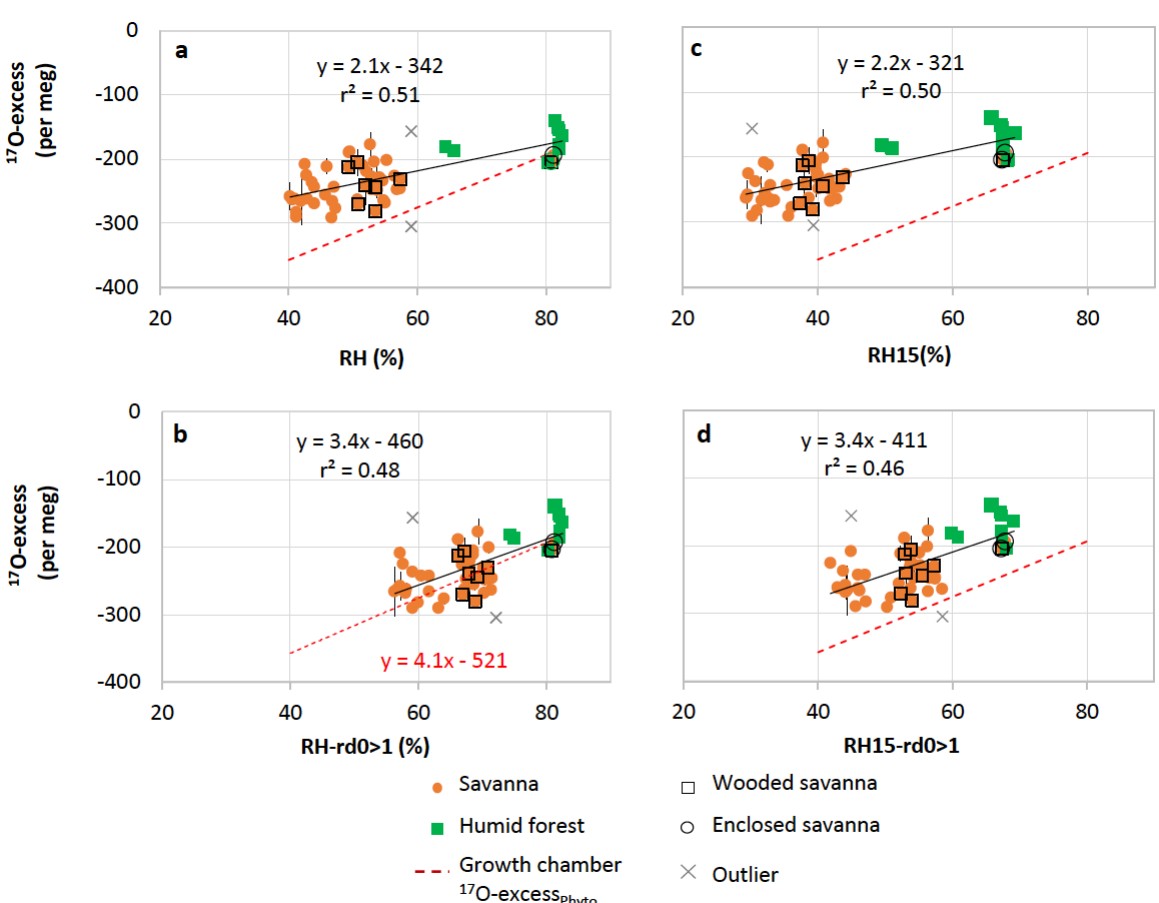

**Figure 5**

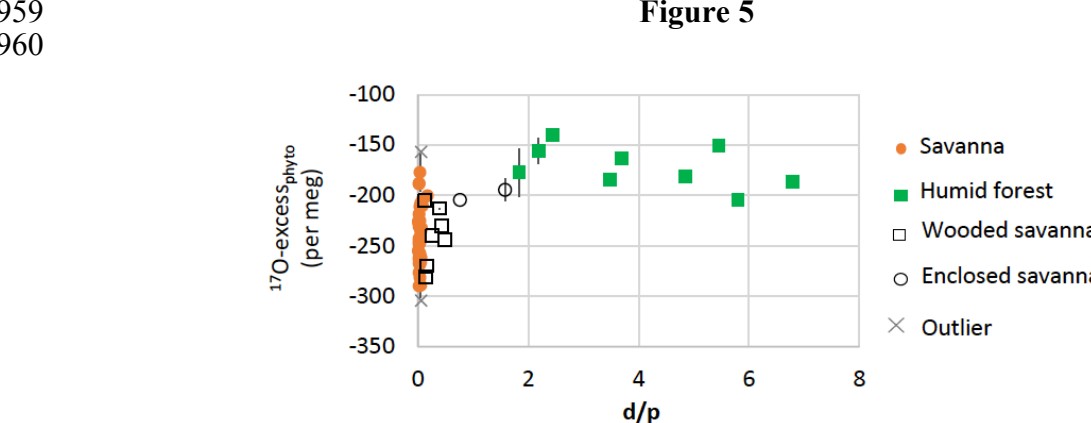

**Figure 6**

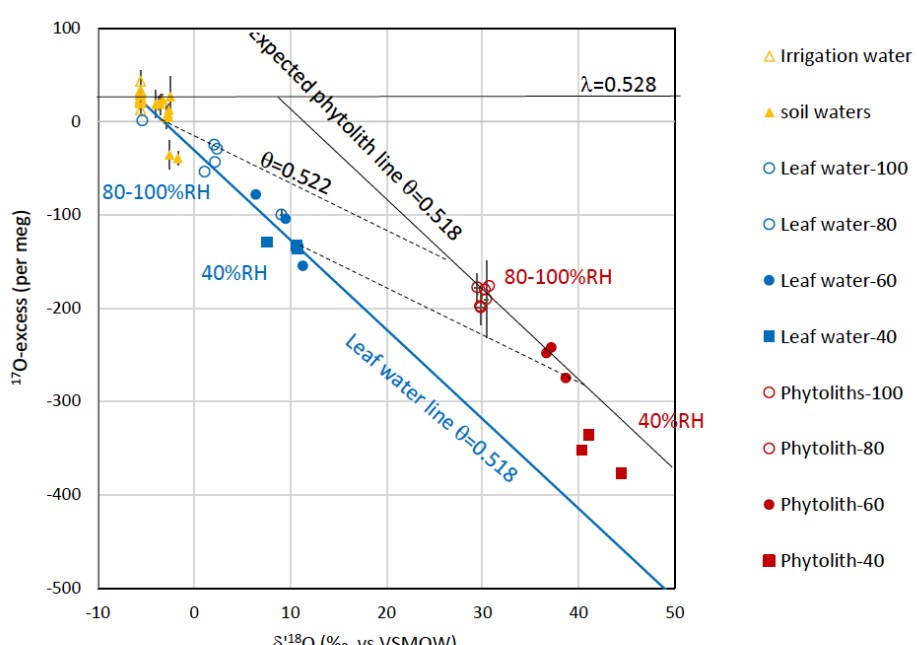

**Table S1: a)** Comparison between IRMS (4 replicates, SD of 0.015‰, 0.010‰ and 5 per meg for $\delta^{17}O$, $\delta^{18}O$ and $^{17}O$-excess respectively) and laser analyser (12 replicates, SD displayed) measurements of working water standards. SD for standard deviation; **b)** Measurements of soil water samples with the isotope laser analyzer (Picarro L2140i) operated in $^{17}O$-excess mode with and without the Picarro micro combustion module (MCM); SD: standard deviation calculated on the replicates.

a)

| | Laser analyzer Picarro L2140i (Ecotron) | | | IRMS MAT 253 (LSCE) | | | Difference laser analyzer/IRMS | | |
|---|---|---|---|---|---|---|---|---|---|
| | $\delta^{18}O$ | $\delta^{17}O$ | $^{17}O$-excess | $\delta^{18}O$ | $\delta^{17}O$ | $^{17}O$-excess | $\delta^{18}O$ | $\delta^{17}O$ | $^{17}O$-excess |
| | ‰ | ‰ | per meg | ‰ | ‰ | per meg | ‰ | ‰ | per meg |
| GIENS-1 | -0.13 | -0.07 | 1.11 | -0.26 | -0.14 | -5.30 | 0.13 | 0.08 | 6.40 |
| ECO-1 | -5.68 | -2.97 | 29.13 | -5.61 | -2.94 | 28.10 | -0.07 | -0.03 | 1.04 |
| ICEBERG-1 | -26.88 | -14.25 | 36.09 | -27.13 | -14.38 | 35.43 | 0.24 | 0.13 | 0.66 |

b)

WITHOUT MCM (3 replicates)   WITH MCM (3 replicates)

| Sample | $\delta^{18}O$ | | | | $\delta^2H$ | | | | $\delta^{17}O$ | | | | $^{17}O$-excess | | | |
|---|---|---|---|---|---|---|---|---|---|---|---|---|---|---|---|---|
| | | SD | | SD | | SD | | SD | | SD | | SD | | SD | | |
| | ‰ | | ‰ | | ‰ | | ‰ | | ‰ | | ‰ | | per meg | | per meg | |
| B3-100-10-05-16 | -2.643 | 0.029 | -2.607 | 0.010 | -18.704 | 0.187 | -18.580 | 0.019 | -1.392 | 0.014 | -1.365 | 0.009 | 4.1 | 3.4 | 12.7 | 5.8 |
| B2-60-10-05-16 | -3.495 | 0.014 | -3.469 | 0.023 | -23.750 | 0.082 | -23.541 | 0.073 | -1.835 | 0.010 | -1.814 | 0.019 | 12.0 | 6.4 | 18.7 | 11.1 |
| B3-100-03-06-16 | -2.799 | 0.018 | -2.766 | 0.022 | -18.868 | 0.105 | -18.894 | 0.185 | -1.462 | 0.022 | -1.457 | 0.019 | 16.7 | 12.7 | 4.8 | 7.4 |
| M1-40-03-06-16 | -5.605 | 0.020 | -5.584 | 0.005 | -31.737 | 0.077 | -31.684 | 0.155 | -2.938 | 0.012 | -2.929 | 0.004 | 25.7 | 3.0 | 23.5 | 1.7 |
| B1-85-10-05-16 | -2.945 | 0.038 | -2.901 | 0.010 | -20.987 | 0.018 | -20.925 | 0.050 | -1.551 | 0.045 | -1.528 | 0.008 | 4.7 | 25.4 | 4.8 | 12.1 |
| B10-40-10-05-16 | -2.726 | 0.029 | -2.697 | 0.022 | -19.891 | 0.071 | -19.594 | 0.097 | -1.434 | 0.030 | -1.416 | 0.015 | 6.8 | 16.2 | 8.5 | 10.3 |
| B1-40-03-06-16 | -3.903 | 0.011 | -3.895 | 0.005 | -25.017 | 0.187 | -24.959 | 0.025 | -2.041 | 0.012 | -2.040 | 0.009 | 21.6 | 6.4 | 18.9 | 10.7 |
| | | 0.023 | | 0.014 | | 0.104 | | 0.086 | | 0.021 | | 0.012 | | 10.5 | | 8.4 |

**Table S2.** Growth chamber experiment : measured $\delta^{18}O$, $\delta^{17}O$ and $^{17}O$-excess of irrigation water (IW), soil water, leaf water (LW) and phytoliths. Av : average ; n : number of replicates ;  SD : standard deviation calculated on the replicates; n.v. : no value.

| Sample | Irrigation water | | | | | | | Soil water | | | | | | | Leaf water | | | | | | | Phytoliths | | | | | | | |
|---|---|---|---|---|---|---|---|---|---|---|---|---|---|---|---|---|---|---|---|---|---|---|---|---|---|---|---|---|---|
| | $\delta^{18}O$ | SD | $\delta^{17}O$ | SD | n | $\delta^{18}O$ | $^{17}O$-excess | $\delta^{18}O$ | SD | $\delta^{17}O$ | SD | n | $\delta^{18}O$ | $^{17}O$-excess | $\delta^{18}O$ | SD | $\delta^{17}O$ | SD | n | $\delta^{18}O$ | $^{17}O$-excess | $\delta^{18}O$ | SD | $\delta^{17}O$ | SD | n | $\delta^{18}O$ | $^{17}O$-excess | SD |
| | ‰ | | ‰ | | | ‰ | per meg | ‰ | | ‰ | | | ‰ | per meg | ‰ | | ‰ | | | ‰ | per meg | ‰ | | ‰ | | | ‰ | per meg | |
| P1-40-29-04-16 | -5.546 | 0.017 | -2.912 | 0.013 | 3 | -5.562 | 20 | -2.562 | 0.026 | -1.389 | 0.029 | 3 | -2.565 | 36 | 10.733 | 0.106 | 5.519 | 0.082 | 2 | 10.676 | -133 | 45.454 | 0.212 | 23.361 | 0.152 | 2 | 44.451 | -378 | 41 |
| P10-40-10-05-16 | -5.594 | 18.139 | -2.933 | 16.016 | 3 | -5.610 | 25 | -2.697 | 0.022 | -1.416 | 0.015 | 3 | -2.701 | 9 | 7.590 | | 3.870 | | 1 | 7.561 | -130 | 41.947 | 0.348 | 21.590 | 0.199 | 2 | 41.091 | -336 | 15 |
| P1-40-20-05-16 | -5.580 | 0.019 | -2.917 | 0.019 | 3 | -5.596 | 33 | -3.658 | 0.013 | -1.913 | 0.013 | 3 | -3.665 | 20 | 10.807 | | 5.554 | | 1 | 10.749 | -137 | 41.150 | 0.592 | 21.161 | 0.291 | 2 | 40.326 | -352 | 18 |
| P1-40-03-06-16 | n.v. | | n.v. | | | n.v. | n.v. | n.v. | | n.v. | | | n.v. | n.v. | 8.530 | | 4.360 | | 1 | 8.494 | -135 | 41.758 | | 21.451 | | 1 | 40.909 | -376 | |
| Av. | | | | | | -5.589 | 26 | | | | | | -2.977 | 21 | | | | | | 9.370 | -134 | | | | | | 41.694 | -360 | |
| SD | | | | | | 0.025 | 6 | | | | | | 0.600 | 14 | | | | | | 1.596 | 3 | | | | | | 1.867 | 20 | |
| P10-60-29-04-16 | -5.564 | 0.007 | -2.929 | 0.008 | 3 | -5.579 | 13 | -2.504 | 0.067 | -1.296 | 0.057 | 3 | -2.507 | 27 | 9.581 | 0.015 | 4.942 | 0.008 | 2 | 9.535 | -104 | 39.426 | 0.528 | 20.346 | 0.255 | 4 | 38.669 | -275 | 23 |
| P2-60-10-05-16 | -5.563 | 0.001 | -2.917 | 0.016 | 3 | -5.579 | 24 | -3.469 | 0.023 | -1.814 | 0.019 | 3 | -3.475 | 19 | 11.370 | | 5.832 | | 1 | 11.306 | -154 | 37.883 | 0.340 | 19.579 | 0.184 | 4 | 37.183 | -243 | 4 |
| P10-60-20-05-16 | -5.566 | 0.021 | -2.920 | 0.027 | 3 | -5.582 | 23 | -3.260 | 0.028 | -1.699 | 0.008 | 3 | -3.266 | 23 | 6.453 | | 3.323 | | 1 | 6.432 | -78 | 37.368 | 0.504 | 19.306 | 0.257 | 2 | 36.687 | -249 | 4 |
| P10-60-03-06-16 | n.v. | | n.v. | | | n.v. | n.v. | n.v. | | n.v. | | | | | 2.488 | | 1.241 | | 1 | 2.485 | -72 | 36.034 | | 18.597 | | 1 | 35.400 | -265 | |
| Av. | | | | | | -5.580 | 20 | | | | | | -3.083 | 23 | | | | | | 7.440 | -102 | | | | | | 36.985 | -258 | |
| SD | | | | | | 0.002 | 6 | | | | | | 0.509 | 4 | | | | | | 3.869 | 37 | | | | | | 1.351 | 15 | |
| P2-85-29-04-16 | -5.594 | 0.014 | -2.937 | 0.001 | 3 | -5.610 | 21 | -1.667 | 0.016 | -0.920 | 0.010 | 3 | -1.668 | 7 | 2.219 | 0.067 | 1.127 | 0.050 | 2 | 2.217 | -44 | 30.718 | 0.385 | 15.920 | 0.212 | 3 | 30.255 | -180 | 7 |
| P1-85-10-05-16 | -5.542 | 22.510 | -2.898 | 22.807 | 3 | -5.558 | 33 | -2.901 | 0.010 | -1.528 | 0.008 | 3 | -2.905 | 5 | 2.402 | | 1.238 | | 1 | 2.399 | -30 | 31.151 | 0.206 | 16.149 | 0.122 | 3 | 30.675 | -176 | 1 |
| P2-85-20-05-16 | -5.561 | 0.014 | -2.897 | 0.018 | 3 | -5.577 | 43 | -3.975 | 0.018 | -2.082 | 0.010 | 3 | -3.983 | 19 | 1.103 | | 0.528 | | 1 | 1.102 | -54 | 30.218 | 0.070 | 15.642 | 0.036 | 2 | 29.770 | -198 | 15 |
| P2-85-03-06-16 | n.v. | | n.v. | | | n.v. | n.v. | n.v. | | | | 3 | | | 0.802 | | 0.391 | | 1 | 0.802 | -32 | 30.134 | 0.252 | 15.552 | 0.090 | 2 | 29.689 | -244 | |
| Av. | | | | | | -5.581 | 32 | | | | | | -2.852 | 9 | | | | | | 1.630 | -40 | | | | | | 30.098 | -199 | |
| SD | | | | | | 0.026 | 11 | | | | | | 1.158 | 8 | | | | | | 0.796 | 11 | | | | | | 0.459 | 31 | |
| P3-100-10-05-16 | -5.582 | 0.034 | -2.930 | 0.028 | 3 | -5.597 | 21 | -2.607 | 0.010 | -1.365 | 0.009 | 3 | -2.611 | 13 | 9.125 | 1.955 | 4.707 | 0.986 | 2 | 9.084 | -100 | 30.876 | 0.027 | 15.992 | 0.003 | 3 | 30.409 | -190 | 17 |
| P3-100-20-05-16 | -5.572 | 10.963 | -2.916 | 6.038 | 3 | -5.588 | 29 | -2.677 | 0.015 | -1.409 | 0.007 | 3 | -2.680 | 6 | 2.121 | | 1.094 | | 1 | 2.119 | -25 | 29.901 | 0.148 | 15.497 | 0.071 | 3 | 29.463 | -178 | 6 |
| P3-100-03-06-16 | n.v. | | n.v. | | | n.v. | n.v. | n.v. | | n.v. | | | n.v. | n.v. | -5.382 | | -2.844 | | 1 | -5.396 | 1 | 30.286 | | 15.676 | | 1 | 29.837 | -199 | |
| Av. | | | | | | -5.593 | 25 | | | | | | -2.646 | 9 | | | | | | 1.935 | -41 | | | | | | 29.903 | -189 | |
| SD | | | | | | 0.007 | 5 | | | | | | 0.049 | 5 | | | | | | 7.242 | 53 | | | | | | 0.477 | 11 | |
| Av.(a) | | | | | | -5.586 | 26 | | | | | | -2.889 | 16 | | | | | | | | | | | | | | | |
| SD (a) | | | | | | 0.006 | 5 | | | | | | 0.188 | 8 | | | | | | | | | | | | | | | |

(a) Calculated on the raw values.

Table S3. Growth chamber experiment: predicted isotopic enrichment in $^{18}O$ from irrigation water to leaf water ($\Delta^{18}_{LW-IW}$) after Cernusak et al. (2016 ; Additional Supporting information). Refer to Cernusak et al. (1996) for symbol and calculations used in the table. Added calculations are displayed in grey columns: $\Delta^{17}_{LW-IW}$ and $^{17}O$-excess$_e$ were calculated using $^{17}\alpha_{eq} = {}^{18}\alpha_{eq}{}^{0.529}$ and $^{17}\alpha_k = {}^{18}\alpha_{eq}{}^{0.518}$, for the equilibrium fractionation and kinetic fractionation, respectively. $\theta_{LW-IW}$ was calculated as defined in the text. IW: irrigation water; LW : leaf water (LW).

| Sample | Air tem. °C | Leaf temp. °C | Air RH % | Stomatal cond. mol m⁻² s⁻¹ | Boundary layer cond. mol m⁻² s⁻¹ | Atm. vapor $\delta^{18}O$ ‰ | Atm. vapor $\delta^{17}O$ ‰ | IW $\delta^{18}O$ ‰ | IW $\delta^{17}O$ ‰ | LW $\delta^{18}O$ ‰ | LW $\delta^{17}O$ ‰ | air vapor pressure $e_a$ kPa | leaf vapor pressure $e_i$ kPa | $w_a/w_i$ | $\varepsilon_k$ for $\delta^{18}O$ ‰ | $\varepsilon_k$ for $\delta^{17}O$ ‰ | $\varepsilon^*$ for $\delta^{18}O$ at leaf temp ‰ | $\varepsilon^*$ for $\delta^{17}O$ at leaf temp ‰ | $\Delta_v$ for $\delta^{18}O$ ‰ | $\Delta_v$ for $\delta^{17}O$ ‰ | $\Delta^{18}_{LW-IW}$ ‰ | $\Delta^{17}_{LW-IW}$ ‰ | Pred. $\Delta'^{18}_{LW-IW}$ ‰ | Pred. $\Delta'^{17}_{LW-IW}$ ‰ | Pred. $^{17}O$-excess$_e$ LW-IW per meg | Pred. $\theta_{LW-IW}$ | Obs. $\Delta^{18}_{LW-IW}$ ‰ | Obs. $\Delta'^{18}_{LW-IW}$ ‰ | Obs. $\Delta'^{17}_{LW-IW}$ ‰ | Obs. $^{17}O$-excess$_e$ LW-IW per meg | Obs. $\theta_{LW-IW}$ |
|---|---|---|---|---|---|---|---|---|---|---|---|---|---|---|---|---|---|---|---|---|---|---|---|---|---|---|---|---|---|---|---|
| P1-40-29-04-16 | 25.0 | 25.0 | 41.2 | 0.031 | 2 | -5.55 | -2.91 | -5.55 | -2.91 | 10.73 | 5.52 | 1.31 | 3.18 | 0.41 | 27.860 | 14.336 | 9.386 | 4.954 | 0.000 | 0.000 | 25.922 | 13.426 | 25.591 | 13.336 | -176 | 0.521 | 16.370 | 16.238 | 8.420 | -154 | 0.519 |
| P10-40-10-05-16 | 25.0 | 25.0 | 41.3 | 0.032 | 2 | -5.59 | -2.93 | -5.59 | -2.93 | 7.59 | 3.87 | 1.31 | 3.18 | 0.41 | 27.860 | 14.336 | 9.386 | 4.954 | 0.000 | 0.000 | 25.893 | 13.411 | 25.564 | 13.322 | -176 | 0.521 | 13.259 | 13.171 | 6.799 | -155 | 0.516 |
| P1-40-20-05-16 | 25.0 | 25.0 | 41.9 | 0.032 | 2 | -5.58 | -2.92 | -5.58 | -2.92 | 10.81 | 5.55 | 1.33 | 3.18 | 0.42 | 27.857 | 14.334 | 9.386 | 4.954 | 0.000 | 0.000 | 25.723 | 13.324 | 25.398 | 13.236 | -174 | 0.521 | 16.479 | 16.345 | 8.460 | -170 | 0.518 |
| P1-40-03-06-16 | 25.0 | 25.0 | 41.4 | 0.032 | 2 | n.v. | n.v. | n.v. | n.v. | 8.53 | 4.36 | 1.32 | 3.18 | 0.41 | 27.860 | 14.336 | 9.386 | 4.954 | n.v. | n.v. |  |  |  |  |  |  |  |  |  |  |  |
| P10-60-29-04-16 | 25.0 | 25.0 | 60.5 | 0.052 | 2 | -5.56 | -2.93 | -5.56 | -2.93 | 9.58 | 4.94 | 1.92 | 3.18 | 0.61 | 27.770 | 14.290 | 9.386 | 4.954 | 0.000 | 0.000 | 20.458 | 10.627 | 20.252 | 10.571 | -122 | 0.522 | 15.230 | 15.115 | 7.864 | -117 | 0.520 |
| P2-60-10-05-16 | 25.0 | 25.0 | 60.2 | 0.052 | 2 | -5.56 | -2.92 | -5.56 | -2.92 | 11.37 | 5.83 | 1.91 | 3.18 | 0.60 | 27.772 | 14.291 | 9.386 | 4.954 | 0.000 | 0.000 | 20.543 | 10.670 | 20.335 | 10.614 | -123 | 0.522 | 17.028 | 16.885 | 8.737 | -178 | 0.517 |
| P10-60-20-05-16 | 25.0 | 25.0 | 60.5 | 0.052 | 2 | -5.57 | -2.92 | -5.57 | -2.92 | 6.45 | 3.32 | 1.92 | 3.18 | 0.61 | 27.770 | 14.290 | 9.386 | 4.954 | 0.000 | 0.000 | 20.458 | 10.627 | 20.252 | 10.571 | -122 | 0.522 | 12.087 | 12.014 | 6.242 | -101 | 0.520 |
| P10-60-03-06-16 | 25.0 | 25.0 | 60.3 | 0.052 | 2 | n.v. | n.v. | n.v. | n.v. | 2.49 | 1.24 | 1.92 | 3.18 | 0.60 | 27.771 | 14.291 | 9.386 | 4.954 | n.v. | n.v. |  |  |  |  |  |  |  |  |  |  |  |
| P2-85-29-04-16 | 25.0 | 25.0 | 80.2 | 0.074 | 2 | -5.59 | -2.94 | -5.59 | -2.94 | 2.22 | 1.13 | 2.55 | 3.18 | 0.80 | 27.680 | 14.244 | 9.386 | 4.954 | 0.000 | 0.000 | 14.918 | 7.789 | 14.808 | 7.758 | -60 | 0.524 | 7.857 | 7.826 | 4.067 | -65 | 0.520 |
| P1-85-10-05-16 | 25.0 | 25.0 | 76.6 | 0.070 | 2 | -5.54 | -2.90 | -5.54 | -2.90 | 2.40 | 1.24 | 2.44 | 3.18 | 0.77 | 27.697 | 14.252 | 9.386 | 4.954 | 0.000 | 0.000 | 15.928 | 8.306 | 15.802 | 8.272 | -72 | 0.523 | 7.989 | 7.957 | 4.139 | -62 | 0.520 |
| P2-85-20-05-16 | 25.0 | 25.0 | 81.5 | 0.075 | 2 | -5.56 | -2.90 | -5.56 | -2.90 | 1.10 | 0.53 | 2.59 | 3.18 | 0.82 | 27.675 | 14.241 | 9.386 | 4.954 | 0.000 | 0.000 | 14.554 | 7.602 | 14.449 | 7.573 | -56 | 0.524 | 6.702 | 6.679 | 3.429 | -97 | 0.513 |
| P2-85-03-06-16 | 25.0 | 25.0 | 82.5 | 0.076 | 2 | n.v. | n.v. | n.v. | n.v. | 0.80 | 0.39 | 2.62 | 3.18 | 0.83 | 27.670 | 14.239 | 9.386 | 4.954 | n.v. | n.v. |  |  |  |  |  |  |  |  |  |  |  |
| *P3-100-10-05-16* | 25.0 | 25.0 | 100.0 | 0.095 | 2 | -5.58 | -2.93 | -5.58 | -2.93 | 9.13 | 4.71 | 3.18 | 3.18 | 1.00 | 27.592 | 14.199 | 9.386 | 4.954 | 0.000 | 0.000 | 9.386 | 4.954 | 9.342 | 4.942 | 9 | 0.529 | 14.789 | 14.681 | 7.630 | -122 | 0.520 |
| P3-100-20-05-16 | 25.0 | 25.0 | 100.0 | 0.095 | 2 | -5.57 | -2.92 | -5.57 | -2.92 | 2.12 | 1.09 | 3.18 | 3.18 | 1.00 | 27.592 | 14.199 | 9.386 | 4.954 | 0.000 | 0.000 | 9.386 | 4.954 | 9.342 | 4.942 | 9 | 0.529 | 7.736 | 7.706 | 4.014 | -54 | 0.521 |
| P3-100-03-06-16 | 25.0 | 25.0 | 100.0 | 0.095 | 2 | n.v. | n.v. | n.v. | n.v. | -5.38 | -2.84 | 3.18 | 3.18 | 1.00 | 27.592 | 14.199 | 9.386 | 4.954 | n.v. | n.v. |  |  |  |  |  |  |  |  |  |  |  |

Stomatal conductance: gs ranges from 0.1 to 0.5 in investigated C3 grasses is lower than 0.2 in C4 grasses. Cf Ocheltree et al., 2012. Here gs is calculated according to Li et al., 2017.

Boundary layer cond: 0.2 to 3 in Li et al., 2017