# Peer review of "The triple oxygen isotope composition of phytoliths as a proxy of continental atmospheric"

_Biogeosciences, 2017_

## Referee Comment (RC1) · Anonymous Referee #1 · 10 Jan 2018

Review of "The triple oxygen isotope composition of phytoliths as a proxy of continental atmospheric humidity: insights from climate chamber and climate transect calibrations" by Alexandre A. et al. BGD.

**General:**

The manuscript deals with the potential of using phytoliths, micrometric amorphous silica particles that form in continuously living plants, for paleoclimate reconstruction of relative humidity changes. These humidity changes are difficult to reconstruct but are highly important to understand the drivers of climate changes on Earth. Studies on phytoliths have be conducted since long in various research fields such as archaeology, biology, plant physiology as well as paleoclimatology. With the significant improvements of online measurement systems – that are much easier to handle – the amorphous silicate analysis have strongly increased. This includes also the isotope determination of the oxygen that is directly attached to the silica, hence the non-exchangeable oxygen. There are several difficulties to circumvent to come up with reliable and meaningful results. One of the most important issues is to remove the exchangeable OH groups of the silica. Recent developments seem to completely overcome this shortcoming and opens up new opportunities for research using stable isotopes, in particular, in view of the newly available measurement techniques for stable oxygen isotopes, including $^{17}O$. Therefore, the presented work is timely by combining laboratory work and field analysis using state-of-the-art measurement techniques. The question addressed with their research, namely to investigate the relation of the $\Delta^{17}O$ to the relative humidity is important since there is hardly any known parameter available that could be used for paleoclimate work. The authors are experienced to work on phytoliths and/or using isotope analysis for their studies. This is again a well taken combination which I very much appreciate.

I suggest accepting this manuscript after addressing the following points:

Major points:

- Since there have been different measurement techniques used to determine the oxygen isotopes, it would be worthwhile in this context to report the comparability of the results mentioned in an additional table ($H_2O$ on Picarro L2140i and $O_2$ converted from $H_2O$ on Delta V mass spectrometer) as well as the measurements done on the Picarro micro combustion module (MCM) in comparison with direct water measurements.
- How was the difference in $\Delta^{17}O$ between Phyto and LW calculated since from Fig. 1a, I am not able to obtain Fig. 1c for this difference? Please check it. This is also in line with the slopes of LW and Phyto vs. RH being different.
- The comparison of the field with the lab results are critical (line 410 to 417), since there is no reason given why we should take the RH only for those months with a limited precipitation. This is in particular important since the $r^2$ values actually decreases when going from RH or RH15 to the range limited by precipitation. This requires further discussion. It is no argument to fit the field data to the lab data just based on a slope measured.
- A weak point is indeed that no water vapor measurements are performed, this is indeed a strong shortcoming because a Picarro L2140i was available for the study. Yet, the authors clearly pointed out the importance to include such measurements in future studies. Was the leaf water measured for dD? If yes, this may help you with the interpretation in that it helps to make reasonable assumptions for the water vapor values.

- Triple isotope comparison of Phyto with RH: It would be nice to distinguish the LW values given in blue for the high RH values (80-100 %) compared to low RH values (40%). This would allow the reader to better follow fig. 6. You may also use ellipses for these clarifications. Same issue with the Phyto values given in red.

Minor points:

- Why do you clean it cryogenically for NF, is NF produced during the fluorination process? How much could it affect the $^{17}O$ and therefore the $\Delta^{17}O$ results?
- You mentioned that you checked the temperature independencies for $^{18}O$ and $^{17}O$ up to 70°C. Please add more information on this issue, because this is important. How have you done it? Wouldn't it be worthwhile to show the experimental results that you have obtained in this paper?
- There is a significant difference of one of the standard material used, i.e. San Carlos Olivine. Whereas Sharp et al. (2016) reported a normalized $\delta^{18}O$ value of 5.3 ‰ and a 17O-excess value of -54 per meg your values were $\delta^{18}O$ SC = 4.949 ± 0.219 ‰ and 17O-excessSC = -49 ± 24 per meg. Why this difference in $\delta^{18}O$? $\delta^{18}O$
- On line 317 you have used ppm to express $\Delta^{17}O$ whereas you have often used per meg, be consistent over the whole manuscript.

**Specific remarks:**

l. 289: were dehydrated… do you mean adsorbed water or interstitial water?

l. 345: Make sure the minus sign is attached to the number.

l. 362 etc.: Make sure that only relevant digits are given for the measurements according to their uncertainty.

l. 366: I suggest changing….withdrawn from the data set… to ….excluded from further calculations…

l. 388: delete 00 prior to the number 2.

l. 538: add space after for

l. 542f: One can expect **that** the isotope composition…

Table 1: Explain P1-40-29-04-16 etc. in the table legend.

Table 2: Legend not consistent with table.

Fig. 1: add x-axis on the top as well for easier readability. Panel c) is not consistent for me since it should be the difference between the measurements shown under panel a). This is not correct for all points. There should be an increase in Phyto-LW. Am I wrong?

Fig. 5: How relevant is this figure?

Fig. 6: explain the different slope and slope ratios used in the figure.

---

## Referee Comment (RC2) · Anonymous Referee #2 · 8 Feb 2018

The authors present the results of triple oxygen isotope measurements of plant silica. The aim of this study is introducing d17O and d18O of phytoliths as proxy for the relative humidity (RH).

The authors conducted laboratory experiments with controlled irrigation water composition, temperature and relative humidity. Data show that the difference in D17O between irrigation water and phytolith changes with RH. That is expected as the kinetic fractionation becomes more important at lower RH. Because kinetic fractionation follows a slope 0.516 (and 0.528 was used as reference line for defining D17O), resultant

D17O values of the phytolith change with RH.

The authors point out that the weakness of this study is the lack of vapor data. That is true and the study would certainly have benefited from such data. The results show that the phytoliths fall on a line about parallel to the water evaporation trend typical of leaf water.

In figure, the lines should go through all data, including the RH = 100 points. If necessary, draw curves. There is no physical reason why the laws of nature stop operating at RH = 85. It is a continuum; possibly with gradually changing mechanisms above RH = 85. That should not be camouflaged in the figure. For readers with a b&w printer only, different symbols would be appropriate to distinguish the different data.

In figure 1, "17O-excess" (top, bottom) is not relative to VSMOW. It is, however, reported relative to a reference line with slope (0.528) and intercept (0). Delete "VSMOW". Also, the $\Delta 18O$ should not be reported relative to VSMOW; it's a difference between $\delta$ values; delete VSMOW here, too.

Line 386ff: Don't give numbers like 27.948 $\pm$ 7.168 ! Give 28 $\pm$ 7. Only report significant number of digits. See also line 405; never give more digits than the uncertainty allows. Change throughout the entire manuscript.

My major criticism on this paper is that is is not evaluated how precise (+/- RH values) the approach is for the reconstruction of the RH. Also, completely missing is a discussion on the heterogeneity of leaf water and the effect on the phytolith composition. Eventually, people will use fossil phytoliths for reconstructing past RH and they will not know from which part of the plant the samples come. After this assessment, come to a decision on whether this proxy works or not (for useful applications). The lack of a quantitative assessment of all uncertainties is a general problem of many proxies.

With some corrections and such a quantitative discussion, the manuscript is surely worth being published in Biogeosciences.

---

## Author Comment (AC1) · 6 Mar 2018

**We would like to thank reviewer #1 for his/her constructive comments. The points of concern are addressed below. In the revised draft found below, changes are highlighted in grey.** Please note that in this revised draft, most recent references were added regarding the state of the art on biomarker isotope composition (section 1), the way RH is taken into account in GCMs (section 1) and the xylem water isotope signature (section 4.2) (Lehmann et al., 2018; Rach et al., 2017; Stevens et al., 2017).

**Major points**

- *Since there have been different measurement techniques used to determine the oxygen isotopes, it would be worthwhile in this context to report the comparability of the results mentioned in an additional table (H2O on Picarro L2140i and O2 converted from H2O on Delta V mass spectrometer) as well as the measurements done on the Picarro micro combustion module (MCM) in comparison with direct water measurements.*

In agreement with this comment, two tables were added in revised supplementary material: Table S1 **a)** Measurement of the water laboratory standards with the laser analyzer Picarro L2140i and the isotope ratio mass-spectrometer MAT 253; **b)** Measurements of soil water samples with the isotope laser analyzer (Picarro L2140i) operated in $^{17}$O-excess mode with and without the Picarro micro combustion module (MCM). The other tables in the supplementary material are re-numbered accordingly.

- *How was the difference in $\Delta^{17}O$ between Phyto and LW calculated since from Fig. 1a, I am not able to obtain Fig. 1c for this difference? Please check it. This is also in line with the slopes of LW and Phyto vs. RH being different.*

Errors were made in the calculations of $^{17}$O-excess$_e$ and $\theta$. This is now corrected in the figures, tables and text. Calculations of $\delta'^{18}$O, $\delta'^{17}$O, $\Delta'^{18}$O $^{17}$O-excess$_e$, $\theta$ and $\lambda$ are now detailed in the introduction section. The text has been changed accordingly. The corrected data are close to the initial data and the interpretation of the data remains unchanged.

- *The comparison of the field with the lab results are critical (line 410 to 417), since there is no reason given why we should take the RH only for those months with a limited precipitation. This is in particular important since the r2 values actually decreases when going from RH or RH15 to the range limited by precipitation. This requires further discussion. It is no argument to fit the field data to the lab data just based on a slope measured.*

There may be a misunderstanding here. As stated in section 2.2, RH-rd0>1 is the averaged RH monthly means for months with at least one day with precipitation higher than 0.1 mm. It was calculated as a proxy of RH during the wet months, likely those of the grass growing season, which explains why the relationship between $^{17}$O-excess$_{phyto}$ and RH is the closest to the growth chamber's one.

For further clarity we add in the revised draft (section 2.2): "RH-rd0>1 was calculated as a proxy of RH during wet months, likely those of the grass growing season".

And in section 3.2: "The relationship obtained between $^{17}$O-excess$_{Phyto}$ and RH-rd0>1 (i.e. RH of the wet months) is the closest to the one obtained between $^{17}$O-excess$_{phyto}$ and RH in the growth chambers (fig. 4b)."

- *A weak point is indeed that no water vapor measurements are performed, this is indeed a strong shortcoming because a Picarro L2140i was available for the study. Yet, the authors clearly pointed out the importance to include such measurements in future studies. Was the leaf water measured for dD? If yes, this may help you with the interpretation in that it helps to make reasonable assumptions for the water vapor values.*

We indeed tried to measure $\delta$D on leaf water using a CRDS analyzer Picarro L2130i. Still, as already demonstrated by previous studies (e.g. Schmidt et al., 2012), because of optical interference, the values are most of the time erroneous. This could be checked by a comparison with the $\delta^{18}$O values produced by fluorination-IRMS showing that half of the CRDS δD values were off. We thus refrain from
interpreting the δD values obtained through this approach.

• *Triple isotope comparison of Phyto with RH: It would be nice to distinguish the LW values given*
*in blue for the high RH values (80-100 %) compared to low RH values (40%). This would allow*
*the reader to better follow fig. 6. You may also use ellipses for these clarifications. Same issue with*
*the Phyto values given in red.*

In agreement with this comment, figure 6 has been redone to differentiate phytolith and plant water data at
80-100%RH, 60%RH and 40%RH.

**Minor points:**

• *Why do you clean it cryogenically for NF, is NF produced during the fluorination process? How*
*much could it affect the 17O and therefore the d17O results?*

We assume that NF may be produced from the fluorination of residual organic N in phytoliths.
However, it is also possible that the interference of the $^4N^9F$ ion on the masse 33 is negligible. As a
matter of fact we could not detect any $^4N^9F_2$ (mass 52) on the mass-spectrometer ThermoQuest Finnigan
Delta Plus when analysing terrestrial or extraterrestrial materials. Some of our internal quartz and
phytolith standards were analysed with and without an extra slush-step. They gave similar results ($^{17}$O-
excess of Boulangé: -0.110±0.031, n=148 without slush step, -0.104±0.022, n=63 with slush step; $^{17}$O-
excess of MSG60: -0.216±0.033, n=22 without slush step, -0.212±0.043, n=7 with slush step).
However, by caution, and in order to follow the same $O_2$ extraction protocol when analyzing terrestrial
quartz, olivine, garnet and phytolith, as well as extra-terrestrial samples, the CEREGE stable isotopes
laboratory chose to keep the slush step. Further comparisons with and without the slush step on several
phytolith samples would be necessary to finally decide if the slush step is useful or not.

The revised draft was modified in section 2.6.1 as follows: "The purified oxygen gas ($O_2$) was passed
through a –114 °C slush to refreeze gases interfering with the mass 33 (e.g. NF), potentially produced
during the fluorination of residual organic N, …"

• *You mentioned that you checked the temperature independencies for 18O and 17O up to 70°C.*
*Please add more information on this issue, because this is important. How have you done it?*
*Wouldn't it be worthwhile to show the experimental results that you have obtained in this paper?*

In agreement with this comment we gave details in the revised draft as follows (section 2.3): "It has
been shown that up to a temperature of 70 °C the extraction has no effect on the $\delta^{18}$O (Crespin et al.,
2008). We verified that it did not have any effect on the $^{17}$O-excess either, using our internal standard
MSG extracted at 60 and 70°C (Crespin et al., 2008). The obtained $^{17}$O-excess values were similar (-
211 and -243 per meg, respectively) given our reproducibility of ±34 per meg (see section 2.6.1)."

The way our internal phytolith standard MSG was extracted from a mascareignite soil sample at
different temperature has been described in details in Crespin et al. (2008) as stated in the manuscript.

• *There is a significant difference of one of the standard material used, i.e. San Carlos Olivine.*
*Whereas Sharp et al. (2016) reported a normalized δ18O value of 5.3 ‰ and a 17O-excess value*
*of -54 per meg your values were δ18O SC = 4.949 ± 0.219 ‰ and 17O-excessSC = -49 ± 24 per*
*meg. Why this difference in δ18O? δ18O*

This point is now discussed in the revised draft as follows (section 2.6.1): "As previously discussed in
Suavet et al. (2010), a large scatter is often observed for SC olivine $\delta^{18}$O and $\delta^{17}$O values measured in a
given laboratory or from a laboratory to another. This is probably attributable to the heterogeneity of the
analyzed samples. At CEREGE, the internal standard of SC olivine is prepared from a number of millimetric
crystals with possibly different oxygen isotope composition. The $\delta^{18}$O and $\delta^{17}$O values from Suavet et al.
(2010), Tanaka and Nakamura (2013) Pack et al. (2016), Sharp et al. (2016) and the present study average

5.295 ± 0.228 (1 SD) ‰ and 2.721 ± 0.121 (1 SD) ‰, respectively. Nevertheless, despite the large SD on
$^{18}O$ and $\delta^{17}O$ measurements, the SC olivine $^{17}O$-excess appears relatively constant (-71 ± 23 (1 SD) per
meg).

• *On line 317 you have used ppm to express   17O whereas you have often used per meg, be*
*consistent over the whole manuscript.*

Corrected

**Specific remarks:**

• *l. 289: were dehydrated…do you mean adsorbed water or interstitial water?*

Corrected: dehydrated and dehydroxylated

• *l. 345: Make sure the minus sign is attached to the number.*

Corrected.

• *l. 362 etc.: Make sure that only relevant digits are given for the measurements according to their*
*uncertainty.*

Three digits of precision on $\delta^{18}O$ and $\delta^{17}O$ values are necessary as 17O-excess is expressed in per meg.

• *l. 366: I suggest changing….withdrawn from the data set… to ….excluded from further*
*calculations…*

Corrected.

• *l. 388: delete 00 prior to the number 2.*

Corrected.

• *l. 538: add space after for*

Corrected.

• *l. 542f: One can expect **that** the isotope composition…*

Corrected.

• *Table 1: Explain P1-40-29-04-16 etc. in the table legend.*

Corrected.

• *Table 2: Legend not consistent with table.*

Corrected.

• *Fig. 1: add x-axis on the top as well for easier readability. Panel c) is not consistent for me since*
*it should be the difference between the measurements shown under panel a). This is not correct for*
*all points. There should be an increase in Phyto-LW. Am I wrong?*

This is right. This was corrected in the revised version. Ses answers to the major points. x-axis is added on
top of fig. A in the revised draft.

• *Fig. 5: How relevant is this figure?*

Fig.5 is not essential but is relevant to discuss the impact of the vegetation source and of the proportion of
the Globular granulate phytoliths (assumed to come from the non-transpiring secondary xylem of the wood)
on the $^{17}O$-excess of phytoliths. This is discussed in section 4.2.

• *Fig. 6: explain the different slope and slope ratios used in the figure.*

For further clarity, this is now explained in caption of figure 6 in the revised draft. The associated paragraph
in the text (section 4.2) was rewritten accordingly.

[revised manuscript text omitted]
 (mm) | MAT (°C) | RH (%) | RH-rd0>1 (%) | RH15 (%) | RH15-rd0>1 (%) | δ'18OPre (1) (‰) | d/p | n | δ'18OPhyto (‰) | SD | δ'17OPhyto (‰) | SD | 17O-excessPhyto (per meg) | SD | δ'18OPhyto FW (‰) | Δ'18OPhyto-Pre-rd0>1 |
|---|---|---|---|---|---|---|---|---|---|---|---|---|---|---|---|---|---|---|---|
| **Savana** | | | | | | | | | | | | | | | | | | | |
| RIM 3 | 21.5 | -13.0 | 52.4 | 27.3 | 47.1 | 61.7 | 35.4 | 47.0 | -3.220 | 0.03 | | 33.127 | | 17.218 | | -243 | | 2.384 | 36.351 |
| RIM 8 | 21.0 | -12.2 | 49.1 | 28.2 | 44.1 | 60.5 | 33.0 | 45.9 | -3.420 | 0.04 | | 34.813 | | 18.304 | | -243 | | 4.221 | 38.239 |
| MAU06 | 20.6 | -12.6 | 68.8 | 27.6 | 44.0 | 58.0 | 33.0 | 44.1 | -3.829 | 0.04 | | 28.871 | | 15.088 | | -268 | | -1.816 | 32.707 |
| RIM 11 | 16.9 | -15.2 | 209.1 | 27.3 | 45.9 | 68.5 | 32.5 | 52.2 | -4.047 | 0.04 | | 37.506 | | 19.785 | | -211 | | 6.745 | 41.561 |
| RIM 10 | 16.7 | -15.2 | 227.6 | 27.2 | 45.7 | 68.7 | 32.1 | 52.1 | -4.042 | 0.01 | | 38.163 | | 20.094 | | -256 | | 7.377 | 42.214 |
| S33 | 16.4 | -14.8 | 270.5 | 27.7 | 42.7 | 57.6 | 29.7 | 41.8 | -3.861 | 0.04 | | 35.961 | | 18.939 | | -225 | | 5.276 | 39.829 |
| S32 | 16.3 | -15.4 | 284.4 | 27.3 | 46.9 | 61.6 | 33.5 | 46.2 | -3.768 | 0.04 | | 37.297 | | 19.617 | | -266 | | 6.537 | 41.072 |
| C4L1 | 16.1 | -14.0 | 287.7 | 29.8 | 40.9 | 57.1 | 29.4 | 42.9 | -3.874 | 0.02 | 2 | 34.915 | 0.368 | 18.340 | 0.203 | -262 | 11 | 4.609 | 38.797 |
| S40 | 16.1 | -13.9 | 329.1 | 29.2 | 40.6 | 56.8 | 29.4 | 43.0 | -3.969 | 0.06 | | 35.385 | | 18.592 | | -262 | | 4.967 | 39.363 |
| S29 | 16.1 | -14.9 | 313.0 | 27.8 | 43.6 | 59.1 | 30.8 | 43.7 | -3.833 | 0.05 | 2 | 35.449 | 0.583 | 18.653 | 0.303 | -236 | 0 | 4.785 | 39.290 |
| 82-46 | 16.0 | -16.0 | 316.4 | 27.1 | 53.0 | 67.5 | 40.1 | 54.2 | -3.604 | 0.03 | | 33.575 | | 17.654 | | -228 | | 2.800 | 37.185 |
| 82-47 | 16.0 | -16.0 | 316.4 | 27.1 | 53.0 | 67.5 | 40.1 | 54.2 | -3.604 | 0.04 | | 36.429 | | 19.169 | | -247 | | 5.642 | 40.039 |
| S44 | 15.8 | -13.5 | 369.1 | 29.6 | 40.2 | 57.2 | 29.6 | 44.1 | -4.073 | 0.04 | 2 | 36.211 | 0.593 | 19.041 | 0.284 | -258 | 24 | 5.863 | 40.292 |
| C4L3 | 15.4 | -13.7 | 467.7 | 29.6 | 41.2 | 59.1 | 30.3 | 45.7 | -4.023 | 0.05 | 2 | 33.688 | 0.312 | 17.652 | 0.175 | -290 | 13 | 3.345 | 37.719 |
| S54 | 15.3 | -13.0 | 443.6 | 29.7 | 41.3 | 60.0 | 31.0 | 47.2 | -4.009 | 0.04 | | 35.586 | | 18.680 | | -282 | | 5.261 | 39.603 |
| S58 | 15.1 | -12.8 | 478.6 | 29.7 | 42.0 | 56.3 | 31.7 | 44.3 | -4.009 | 0.05 | 2 | 36.161 | 0.234 | 19.006 | 0.143 | -266 | 21 | 5.833 | 40.179 |
| C5L1 | 15.0 | -12.9 | 583.2 | 29.7 | 42.5 | 57.2 | 32.1 | 44.9 | -3.972 | 0.06 | 3 | 29.525 | 0.483 | 15.500 | 0.257 | -208 | 7 | -0.787 | 33.505 |
| 83-62 | 14.9 | -12.3 | 515.8 | 29.7 | 42.9 | 58.1 | 32.6 | 46.0 | -4.097 | 0.06 | 2 | 36.320 | 0.747 | 19.095 | 0.424 | -262 | 36 | 5.987 | 40.426 |
| S5 | 14.7 | -16.2 | 511.1 | 28.1 | 53.3 | 68.6 | 39.2 | 53.9 | -3.789 | 0.08 | 2 | 24.297 | 0.115 | 12.704 | 0.064 | -205 | 4 | -6.312 | 28.094 |
| 82-79 | 14.2 | -16.1 | 669.0 | 28.3 | 54.2 | 70.1 | 39.9 | 55.2 | -3.774 | 0.03 | 2 | 33.913 | 0.046 | 17.798 | 0.076 | -229 | | 3.356 | 37.694 |
| 83-75 | 14.1 | -12.7 | 736.2 | 29.1 | 46.7 | 63.1 | 35.6 | 50.3 | -3.936 | 0.03 | | 32.418 | | 16.969 | | -290 | | 2.000 | 36.362 |
| 82-78 | 14.1 | -16.1 | 669.0 | 28.3 | 55.2 | 71.0 | 40.8 | 56.2 | -3.768 | 0.18 | | 23.789 | | 12.437 | | -201 | | -6.785 | 27.565 |
| S84 | 13.9 | -13.4 | 775.2 | 28.9 | 47.4 | 64.1 | 36.1 | 50.9 | -4.040 | 0.03 | 2 | 32.600 | 0.435 | 17.080 | 0.221 | -277 | 5 | 2.141 | 36.648 |
| S118 | 13.6 | -13.7 | 878.1 | 28.6 | 49.6 | 66.3 | 37.7 | 52.9 | -4.008 | 0.02 | | 30.007 | | 15.779 | | -188 | | -0.501 | 34.023 |
| S88 | 13.6 | -13.6 | 880.0 | 28.6 | 49.4 | 66.2 | 37.7 | 52.9 | -3.996 | 0.02 | | 28.371 | | 14.900 | | -189 | | -2.129 | 32.375 |
| 83-120 | 13.5 | -13.8 | 934.5 | 28.5 | 50.7 | 67.3 | 38.7 | 53.8 | -3.984 | 0.05 | | 31.622 | | 16.570 | | -262 | | 1.101 | 35.614 |
| 83-122 | 13.4 | -14.9 | 947.3 | 28.1 | 53.6 | 68.1 | 39.8 | 53.7 | -3.928 | 0.03 | 2 | 31.240 | 0.628 | 16.396 | 0.335 | -231 | 9 | 0.649 | 35.176 |
| S122 | 13.3 | -13.9 | 934.5 | 28.5 | 52.1 | 68.1 | 39.7 | 53.7 | -3.971 | 0.03 | | 34.379 | | 18.095 | | -219 | | 3.851 | 38.350 |
| S93 | 13.3 | -13.2 | 1005.3 | 28.6 | 51.6 | 68.2 | 39.7 | 55.1 | -3.925 | 0.08 | | 30.064 | | 15.787 | | -211 | | -0.435 | 33.989 |
| 83-98 | 13.1 | -12.8 | 1067.0 | 28.7 | 52.7 | 69.3 | 40.8 | 56.3 | -4.060 | 0.04 | | 29.692 | | 15.621 | | -177 | | -0.800 | 33.753 |
| S128 | 13.0 | -14.1 | 1055.1 | 28.2 | 54.7 | 70.2 | 41.7 | 56.5 | -3.765 | 0.07 | 2 | 34.078 | 0.567 | 17.919 | 0.265 | -233 | 29 | 3.500 | 37.843 |
| S.130 | 12.9 | -14.2 | 1113.9 | 28.0 | 55.1 | 70.3 | 41.8 | 56.4 | -3.961 | 0.03 | 3 | 35.909 | 0.515 | 18.692 | 0.254 | -268 | 19 | 5.286 | 39.870 |
| 83-103 | 12.9 | -12.4 | 1114.0 | 28.5 | 53.7 | 70.4 | 41.8 | 56.7 | -4.329 | 0.03 | | 30.499 | | 15.855 | | -249 | | -0.024 | 34.828 |
| S.138 | 12.9 | -14.9 | 1127.1 | 27.7 | 56.8 | 70.9 | 42.4 | 56.6 | -4.069 | 0.03 | 2 | 35.822 | 0.5 | 18.667 | 0.3 | -247 | 5 | 5.138 | 39.891 |
| S136 | 12.8 | -14.7 | 1113.4 | 27.8 | 57.4 | 71.5 | 43.3 | 57.4 | -4.023 | 0.02 | | 33.422 | | 17.355 | | -246 | | 2.767 | 37.445 |
| 83-116 | 12.7 | -12.2 | 1233.2 | 28.5 | 54.8 | 71.4 | 42.7 | 58.5 | -4.316 | 0.03 | 2 | 31.084 | | 16.149 | | -264 | | 0.558 | 35.401 |
| 83-115 | 12.4 | -12.3 | 1301.3 | 27.8 | 56.5 | 66.8 | 44.2 | 53.8 | -4.170 | 0.01 | 2 | 31.524 | 0.4 | 16.418 | 0.2 | -226 | 17 | 0.887 | 35.694 |
| **Wooded savana** | | | | | | | | | | | | | | | | | | | |
| 83-8 | 14.9 | -15.9 | 485.2 | 28.0 | 50.9 | 67.0 | 37.3 | 52.4 | -3.948 | 0.16 | | 36.813 | | 19.167 | | -270 | | 6.181 | 40.762 |
| S7 | 14.8 | -16.0 | 513.6 | 28.2 | 52.0 | 67.9 | 38.0 | 53.1 | -4.263 | 0.26 | | 29.491 | | 15.331 | | -241 | | -1.076 | 33.755 |
| 83-4 | 14.7 | -16.5 | 539.7 | 27.0 | 57.4 | 70.9 | 43.7 | 57.3 | -3.821 | 0.43 | | 26.127 | | 13.565 | | -231 | | -4.665 | 29.948 |
| 82-77 | 14.6 | -16.3 | 535.5 | 28.0 | 53.5 | 69.0 | 39.2 | 54.0 | -3.798 | 0.14 | 2 | 35.214 | 0.8 | 18.312 | 0.4 | -281 | 10 | 4.601 | 39.012 |
| S91 | 13.6 | -13.4 | 883.1 | 28.7 | 49.4 | 66.2 | 37.8 | 53.8 | -3.984 | 0.40 | 2 | 34.512 | 0.2 | 18.009 | 0.1 | -213 | 30 | 4.013 | 38.496 |
| C4L8 | 13.5 | -13.7 | 878.1 | 28.6 | 50.7 | 67.3 | 38.7 | 53.8 | -3.984 | 0.13 | | 32.302 | | 16.850 | | -206 | | 1.793 | 36.286 |
| 83-127 | 13.1 | -14.1 | 1055.1 | 28.2 | 53.6 | 69.3 | 40.7 | 55.5 | -3.785 | 0.49 | 2 | 35.638 | 0.2 | 18.573 | 0.1 | -244 | 1 | 5.054 | 39.423 |
| **Enclosed savana** | | | | | | | | | | | | | | | | | | | |
| Biendi 1 | -2.0 | 11.1 | 1839.0 | 25.9 | 80.9 | 80.9 | 67.4 | 67.4 | -3.687 | 0.76 | | 33.086 | 0.0 | 17.233 | 0.0 | -205 | 20 | 2.096 | 36.773 |
| Doubou | -1.8 | 10.9 | 1986.0 | 25.9 | 81.2 | 81.2 | 67.9 | 67.9 | -3.631 | 1.58 | | 31.931 | 0.6 | 16.665 | 0.3 | -194 | 8 | 0.954 | 35.562 |
| **Humid forest** | | | | | | | | | | | | | | | | | | | |
| 83-151 | 12.5 | -16.6 | 1428.6 | 26.5 | 65.8 | 74.9 | 51.0 | 60.8 | -3.787 | 6.78 | | 33.097 | | 17.288 | | -187 | | 2.221 | 36.884 |
| S155 | 12.5 | -16.3 | 1352.6 | 27.0 | 64.4 | 74.3 | 49.4 | 59.9 | -3.777 | 4.85 | 2 | 29.092 | 0.2 | 15.180 | 0.1 | -181 | 4 | -1.698 | 32.869 |
| 04-94 | | 13.1 | 1676.4 | 24.3 | 81.4 | 81.4 | 65.8 | 65.8 | -4.464 | 2.44 | 2 | 32.638 | 0.2 | 17.093 | 0.1 | -140 | 4 | 1.371 | 37.102 |
| 04-88 | | 12.4 | 1707.0 | 24.9 | 81.9 | 81.9 | 67.1 | 67.1 | -4.458 | 5.45 | | 33.137 | 0.4 | 17.345 | 0.2 | -151 | 18 | 1.977 | 37.595 |
| 04-47 | -0.2 | 12.3 | 1724.0 | 26.2 | 82.1 | 82.1 | 67.4 | 67.4 | -1.515 | 3.48 | | 32.953 | 0.0 | 17.215 | 0.0 | -185 | 17 | 2.026 | 34.468 |
| 04-66 | -0.2 | 12.5 | 1690.6 | 25.8 | 82.0 | 82.0 | 67.3 | 67.3 | -4.354 | 1.84 | | 29.959 | 0.6 | 15.641 | 0.3 | -177 | 24 | -1.040 | 34.314 |
| 04-65 | -0.2 | 12.6 | 1690.6 | 25.8 | 82.0 | 82.0 | 67.3 | 67.3 | -4.195 | 2.19 | | 32.791 | 0.6 | 17.158 | 0.3 | -156 | 25 | 1.791 | 36.985 |
| 04-118 | -0.2 | 10.5 | 2148.4 | 26.4 | 82.5 | 82.5 | 69.2 | 69.2 | -3.556 | 3.69 | | 31.840 | 0.4 | 16.648 | 0.2 | -164 | 4 | 0.945 | 35.396 |
| Dimoniki | -4.1 | 12.4 | 1286.6 | 24.7 | 80.3 | 80.3 | 68.1 | 68.1 | -4.284 | 5.80 | | 30.928 | | 16.123 | | -205 | | -0.275 | 35.212 |
| **Outliers** | | | | | | | | | | | | | | | | | | | |
| RIM1 | 16.7 | -16.0 | 216.4 | 26.7 | 52.3 | 72.2 | 39.4 | 58.6 | -3.857 | 0.06 | | 38.131 | | 19.828 | | -305 | | 7.264 | 41.987 |
| C3L4 | 15.6 | -14.2 | 362.0 | 29.3 | 41.8 | 59.1 | 30.3 | 45.0 | -3.968 | 0.06 | | 25.185 | 0.2 | 13.141 | 0.1 | -157 | 10 | -5.211 | 29.153 |

(1) Amount weighted average for months with at least one day with precipitation>0.1mm

**Figure captions**

**Figure 1.** Growth chamber experiment: a) $^{17}$O-excess *vs* relative humidity (RH) of irrigation water (IW), soil water (SW), leaf water (LW) and phytolith (Phyto). Error bars show standard deviation (SD) on the replicates. They are smaller than the symbol when not shown. b) $^{18}$O-enrichment from irrigation water to leaf water ($\Delta'^{18}O_{LW-IW}$), from irrigation water to phytolith ($\Delta'^{18}O_{Phyto-IW}$) and from leaf water to phytolith ($\Delta'^{18}O_{Phyto-LW}$). c) $^{17}$O-excess associated with the enrichment from irrigation water to leaf water ($^{17}O-excess_{e\ LW-IW}$), from irrigation water to phytolith ($^{17}O-excess_{e\ Phyto-IW}$), and from leaf water to phytolith ($^{17}O-excess_{e\ Phyto-LW}$). d, e and f) linear correlations for the 40-80% RH range extracted from a, b and c, respectively.

**Figure 2.** Growth chamber experiment: phytolith types extracted from *Festuca arundinaceae* and observed in natural light microscopy: epidermal long cell (LC), epidermal short cell (SC).

**Figure 3.** Natural West and Central African transect: $\delta'^{18}O$ of phytoliths ($\delta'^{18}O_{Phyto}$) *vs* relative humidity RH-rd0>1 (see fig. 4 for explanation). Error bars show standard deviation (SD) on the replicates. When not shown, they are smaller than the symbol.

**Figure 4.** Natural West and Central African transect: $^{17}$O-excess *vs* relative humidity (RH) of phytolith assemblages from soil tops collected under savanna, wooded savanna, humid forest and enclosed savanna along a humidity gradient (Table 1). The growth chamber $^{17}O-excess_{Phyto}$ *vs* RH correlation line is displayed for comparison. a) RH-Av: yearly average of monthly means; b) RH-rd0>1: yearly average of monthly means for months with at least one day with precipitation higher than 0.1mm; c) RH15: RH at 15:00 H UTC; d) RH15-rd0>1: RH-rd0>1 at 15:00 H UTC.

**Figure 5.** Natural West and Central African transect: $^{17}$O-excess of phytoliths ($^{17}O-excess_{Phyto}$) *vs* d/p.

**Figure 6.** Growth chamber experiment: $^{17}$O-excess *vs* $\delta'^{18}O$ of irrigation water (IW), soil water (SW), bulk leaf water (LW) and phytolith (Phyto). Error bars show standard deviation (SD) on the replicates. The leaf water line (blue) represents how the triple oxygen isotope composition of the bulk leaf water of *Festuca arundinacea* evolves from an irrigation water signature to a more evaporated water signature when RH decreases. This evolution follows a slope equivalent to $\theta$=0.518 in a $\Delta'^{17}O$ vs $\Delta'^{18}O$ space (table 1). Assuming that phytoliths precipitate from the bulk leaf water, the expected phytolith line (black) should be parallel to the leaf water line as the equilibrium fractionation between phytolith and leaf water is constant at constant temperature (25°C). In the investigated case this fractionation, represented by the black dotted line, is equivalent to $\theta$=0.522 (table 1). The isotope signature of phytoliths formed at RH higher than 40% follow the expected phytolith line. However, the isotope signature of phytoliths formed at 40% RH suggest a forming water more evaporated than the bulk leaf water.

**Figure 1**

[Figure]

**Figure 2**

[Figure]

**Figure 3**

[Figure]

**Figure 4**

[Figure]

**Figure 5**

[Figure]

**Figure 6**

[Figure]

Table S1: a) Measurement of the water laboratory standards with the laser analyzer Picarro L2140i and the isotope ratio mass-spectrometer MAT 253; b) Measurements of soil water samples with the isotope laser analyzer (Picarro L2140i) operated in $^{17}$O-excess mode with and without the Picarro micro combustion module (MCM); SD : standard deviation calculated on the replicates.

a)

| | Laser analyzer Picarro L2140i (Ecotron) | | | IRMS MAT 253 (LSCE) | | | Difference laser analyzer/IRMS | | |
|---|---|---|---|---|---|---|---|---|---|
| | $\delta^{18}$O | $\delta^{17}$O | $^{17}$O-excess | $\delta^{18}$O | $\delta^{17}$O | $^{17}$O-excess | $\delta^{18}$O | $\delta^{17}$O | $^{17}$O-excess |
| | ‰ | ‰ | per meg | ‰ | ‰ | per meg | ‰ | ‰ | per meg |
| GIENS-1 | -0.13 | -0.07 | 1.11 | -0.26 | -0.14 | -5.30 | 0.13 | 0.08 | 6.40 |
| ECO-1 | -5.68 | -2.97 | 29.13 | -5.61 | -2.94 | 28.10 | -0.07 | -0.03 | 1.04 |
| ICEBERG-1 | -26.88 | -14.25 | 36.09 | -27.13 | -14.38 | 35.43 | 0.24 | 0.13 | 0.66 |

b)

[revised manuscript text omitted]

Stomatal conductance: gs ranges from 0.1 to 0.5 in investigated C3 grasses is lower than 0.2 in C4 grasses. Cf Ocheltree et al., 2012. Here gs is calculated according to Liet al., 2017.

Boundary layer cond: 0.2 to 3 in Li et al., 2017

---

## Author Comment (AC2) · 6 Mar 2018

**We would like to thank reviewer #2 for his/her constructive comments. The points of concern are addressed below. In the revised draft, changes are highlighted in grey.**

*The authors present the results of triple oxygen isotope measurements of plant silica. The aim of this study is introducing d17O and d18O of phytoliths as proxy for the relative humidity (RH). The authors conducted laboratory experiments with controlled irrigation water composition, temperature and relative humidity. Data show that the difference in D17O between irrigation water and phytolith changes with RH. That is expected as the kinetic fractionation becomes more important at lower RH. Because kinetic fractionation follows a slope 0.516 (and 0.528 was used as reference line for defining D17O), resultant D17O values of the phytolith change with RH.*

*The authors point out that the weakness of this study is the lack of vapor data. That s true and the study would certainly have benefited from such data. The results show that the phytoliths fall on a line about parallel to the water evaporation trend typical of leaf water.*

- *In figure, the lines should go through all data, including the RH = 100 points. If necessary, draw curves. There is no physical reason why the laws of nature stop operating at RH = 85. It is a continuum; possibly with gradually changing mechanisms above RH = 85. That should not be camouflaged in the figure.*

In agreement with this comment, modifications were made to figure 1 as well as in the text (section 3.1) of the revised draft.

- *For readers with a b&w printer only, different symbols would be appropriate to distinguish the different data.*

This was modified in the revised draft.

- *In figure 1, "17O-excess" (top, bottom) is not relative to VSMOW. It is, however, reported relative to a reference line with slope (0.528) and intercept (0). Delete "VSMOW". Also, the $\Delta$ 18O should not be reported relative to VSMOW; it's a difference between $\delta$ values; delete VSMOW here, too.*

This is right. This was modified in the revised draft.

- *Line 386ff: Don't give numbers like 27.948±7.168 ! Give 28±7. Only report significant number of digits. See also line 405; never give more digits than the uncertainty allows. Change throughout the entire manuscript.*

The precision on $\delta^{17}O$ and $\delta^{18}O$ should indeed be given with only 2 digits. Still, as shown in Landais et al. (2006) for leaf water, the uncertainties on $\delta^{17}O$ and $\delta^{18}O$ are not independent so that the final uncertainty on $^{17}O$-excess should not be calculated from uncorrelated uncertainties on $\delta^{17}O$ and $\delta^{18}O$. This is the reason why, there is a need to keep the 3 digits to properly calculate the $^{17}O$-excess. For further clarity 2 digits are presented in the text and 3 digits are kept in tables.

- *My major criticism on this paper is that it is not evaluated how precise (+/- RH values) the approach is for the reconstruction of the RH.*

In agreement with this comment, this is now discussed in the revised draft (section 4.2):

"Without taking into account the two outliers, the linear regression between RH-rd0>1 and $^{17}O$-excess$_{phyto}$ for a 95% confidence interval can be expressed as follows:

RH-rd0>1 = 0.14 ± 0.02 (S.E) x $^7O$-excess$_{phyto}$ + 100.5 ± 4.7 (S.E)                Eq. 2

where $^{17}O$-excess$_{phyto}$ is expressed in per meg and RH in %, $r^2$ = 0.48, p < 0.001 and S.E. stands for standard error. The S.E. of the predicted RH-rd0>1value is ± 5.6%. However, the data scattering (fig. 4) call for assessing additional parameters that can contribute to changes in $^{17}$O-excess$_{Phyto}$, beside RH,
before using the $^{17}$O-excess$_{phyto}$ for quantitative RH reconstruction."

• *Also, completely missing is a discussion on the heterogeneity of leaf water and the effect on the*
*phytolith composition. Eventually, people will use fossil phytoliths for reconstructing past RH and*
*they will not know from which part of the plant the samples come.*

This is now discussed in section 4.2: "In grasses, leaf water is expected to be more prone to evaporative
enrichment than stem water, and inside the leaf itself, the heterogeneity of evaporative sites repartition and
water movements can lead to a significant heterogeneity in the $\delta^{18}$O signatures of water and phytoliths
(Cernusak et al., 2016; Helliker and Ehleringer, 2000; Webb and Longstaffe, 2002). Soil top phytolith
assemblages likely record several decades of annual phytolith production and their isotope composition is
expected to be an average. This would explain the consistency of the $^{17}$O-excess$_{Phyto}$ data obtained from
bulk grass from climate chambers and bulk grasses from natural savannas. However, further investigation
on the extent of the heterogeneity of $^{17}$O-excess signature of water and phytoliths in mature grasses would
help to clarify the links between water and phytolith signatures and better understand the phytolith proxy."

• *After this assessment, come to a decision on whether this proxy works or not (for useful*
*applications). The lack of a quantitative assessment of all uncertainties is a general problem of*
*many proxies. With some corrections and such a quantitative discussion, the manuscript is surely*
*worth being published in Biogeosciences.*

We agree with all the points of concern raised above (see answers above). However, as written in the
discussion and conclusion sections we would like to emphasize that this is a first step in the assessment and
that complementary calibration steps are required to bring us to an accurate quantitative proxy. These
assessments, described in the discussion section, are in progress.

**References**

Cernusak, L.A., Barbour, M.M., Arndt, S.K., Cheesman, A.W., English, N.B., Feild, T.S., Helliker, B.R.,
Holloway-Phillips, M.M., Holtum, J.A.M., Kahmen, A., et al. (2016). Stable isotopes in leaf water of
terrestrial plants. Plant Cell Environ. *39*, 1087–1102.
Crespin, J., Alexandre, A., Sylvestre, F., Sonzogni, C., Paillès, C., and Garreta, V. (2008). IR laser
extraction technique applied to oxygen isotope analysis of small biogenic silica samples. Anal. Chem.
*80*, 2372–2378.
Franchi, I.A., Wright, I.P., Sexton, A.S., and Pillinger, C.T. (1999). The oxygen-isotopic composition of
Earth and Mars. Meteorit. Planet. Sci. *34*, 657–661.
Helliker, B.R., and Ehleringer, J.R. (2000). Establishing a grassland signature in veins: 18O in the leaf
water of C3 and C4 grasses. Proc. Natl. Acad. Sci. U. S. A. *97*, 7894–7898.
Pack, A., Tanaka, R., Hering, M., Sengupta, S., Peters, S., and Nakamura, E. (2016). The oxygen isotope
composition of San Carlos olivine on the VSMOW2-SLAP2 scale. Rapid Commun. Mass Spectrom.
*30*, 1495–1504.
Schmidt, M., Maseyk, K., Lett, C., Biron, P., Richard, P., Bariac, T., and Seibt, U. (2012). Reducing and
correcting for contamination of ecosystem water stable isotopes measured by isotope ratio infrared
spectroscopy. Rapid Commun. Mass Spectrom. *26*, 141–153.
Sharp, Z.D., Gibbons, J.A., Maltsev, O., Atudorei, V., Pack, A., Sengupta, S., Shock, E.L., and Knauth,
L.P. (2016). A calibration of the triple oxygen isotope fractionation in the SiO2–H2O system and
applications to natural samples. Geochim. Cosmochim. Acta *186*, 105–119.
Suavet, C., Alexandre, A., Franchi, I.A., Gattacceca, J., Sonzogni, C., Greenwood, R.C., Folco, L., and
Rochette, P. (2010). Identification of the parent bodies of micrometeorites with high-precision oxygen
isotope ratios. Earth Planet. Sci. Lett. *293*, 313–320.
Tanaka, R., and Nakamura, E. (2013). Determination of 17O-excess of terrestrial silicate/oxide minerals with respect to Vienna Standard Mean Ocean Water (VSMOW). Rapid Commun. Mass Spectrom.
RCM *27*, 285–297.

[revised manuscript text omitted]
 (day) | Temp. (°C) | SD | RH (%) | SD | Light (mmol/m²/sec) | Transp. (l/day) | Biomass (g) | Sample | Conc. (% d.w.) | LC (%) | LW-IW Δ'¹⁸O (‰) | LW-IW Δ'¹⁷O (‰) | LW-IW ¹⁷O-exⁱₑ (per meg) | LW-IW θ | Phyto-LW Δ'¹⁸O (‰) | Phyto-LW Δ'¹⁷O (‰) | Phyto-LW ¹⁷O-exₑ (per meg) | Phyto-LW θ | Phyto-IW Δ'¹⁸O (‰) | Phyto-IW Δ'¹⁷O (‰) | Phyto-IW ¹⁷O-exₑ (per meg) | Phyto-IW θ |
|---|---|---|---|---|---|---|---|---|---|---|---|---|---|---|---|---|---|---|---|---|---|---|
| 11 | 25 | 0.2 | 41.2 | 1 | 278 |  | 13 | P1-40-29-04-16 | n.v. |  | 16.238 | 8.420 | -154 | 0.519 | 33.776 | 17.589 | -244 | 0.521 | 50.013 | 26.009 | -398 | 0.520 |
| 10 | 25 | 0.2 | 41.3 | 1.1 | 278 | 0.49 | 21 | P10-40-10-05-16 | 0.8 |  | 13.171 | 6.799 | -155 | 0.516 | 33.530 | 17.498 | -206 | 0.522 | 46.701 | 24.297 | -361 | 0.520 |
| 11 | 25 | 0.4 | 41.9 | 1 | 311 | 0.69 | 37 | P1-40-20-05-16 | 0.8 | 21 | 16.345 | 8.460 | -170 | 0.518 | 29.577 | 15.401 | -216 | 0.521 | 45.922 | 23.861 | -385 | 0.520 |
| 14 | 25 | 0.2 | 41.4 | 0.9 | 278 | 0.65 | 38 | P1-40-03-06-16 | 1.8 |  | n.v. | n.v. | n.v. | n.v. | 32.415 | 16.874 | -241 | 0.521 | n.v. | n.v. | n.v. | n.v. |
|  |  |  |  |  |  | **Av.** 0.61 |  |  | **1.2** |  | **15.251** | **7.893** | **-159** | **0.517** | **32.324** | **16.840** | **-227** | **0.521** | **47.545** | **24.723** | **-381** | **0.520** |
|  |  |  |  |  |  | **SD** 0.11 |  |  | 0.6 |  | 1.802 | 0.947 | 9 | 0.001 | 1.925 | 1.011 | 19 | 0.0006 | 2.172 | 1.135 | 19 | 0.0003 |
| 11 | 25 | 0.5 | 60.2 | 2.5 | 311 |  | 21 | P10-60-29-04-16 | n.v. |  | 15.115 | 7.864 | -117 | 0.520 | 29.133 | 15.211 | -171 | 0.522 | 44.248 | 23.075 | -288 | 0.521 |
| 11 | 25 | 0.2 | 60.5 | 1 | 289 | 0.57 | 33 | P2-60-10-05-16 | 0.7 |  | 16.885 | 8.737 | -178 | 0.517 | 25.877 | 13.575 | -88 | 0.525 | 42.761 | 22.312 | -266 | 0.522 |
| 10 | 25 | 0.8 | 60.2 | 4.8 | 311 | 0.60 | 48 | P10-60-20-05-16 | 0.8 | 13 | 12.014 | 6.242 | -101 | 0.520 | 30.254 | 15.804 | -170 | 0.522 | 42.268 | 22.047 | -271 | 0.522 |
| 14 | 25 | 0.6 | 60.3 | 3.2 | 311 | 0.76 | 60 | P10-60-03-06-16 | 1.3 |  | n.v. | n.v. | n.v. | n.v. | 32.915 | 17.186 | -193 | 0.522 | n.v. | n.v. | n.v. | n.v. |
|  |  |  |  |  |  | **Av.** 0.64 |  |  | **0.9** |  | **14.671** | **7.614** | **-132** | **0.519** | **29.545** | **15.444** | **-156** | **0.523** | **43.093** | **22.478** | **-275** | **0.522** |
|  |  |  |  |  |  | **SD** 0.10 |  |  | 0.3 |  | 2.465 | 1.266 | 41 | 0.001 | 2.915 | 1.496 | 46 | 0.0012 | 1.031 | 0.534 | 11 | 0.0001 |
| 11 | 25 | 0.2 | 80.2 | 2.8 | 289 |  | 24 | P2-85-29-04-16 | n.v. |  | 7.826 | 4.067 | -65 | 0.520 | 28.039 | 14.668 | -136 | 0.523 | 35.865 | 18.736 | -201 | 0.522 |
| 10 | 25 | 0.2 | 81.5 | 1.3 | 289 | 0.28 | 27 | P1-85-10-05-16 | 0.4 |  | 7.957 | 4.139 | -62 | 0.520 | 28.276 | 14.783 | -147 | 0.523 | 36.233 | 18.922 | -209 | 0.522 |
| 11 | 25 | 0.2 | 76.6 | 2.5 | 278 | 0.22 | 27 | P2-85-20-05-16 | 0.6 | 10 | 6.679 | 3.429 | -97 | 0.513 | 28.668 | 14.993 | -144 | 0.523 | 35.347 | 18.422 | -241 | 0.521 |
| 14 | 25 | 0.2 | 82.5 | 1.1 | 289 | 0.36 | 37 | P2-85-03-06-16 | 1.0 |  | n.v. | n.v. | n.v. | n.v. | 28.888 | 15.041 | -212 | 0.521 | n.v. | n.v. | n.v. | n.v. |
|  |  |  |  |  |  | **Av.** 0.29 |  |  | **0.7** |  | **7.487** | **3.879** | **-75** | **0.518** | **28.468** | **14.871** | **-160** | **0.522** | **35.815** | **18.694** | **-217** | **0.522** |
|  |  |  |  |  |  | **SD** 0.07 |  |  | 0.3 |  | 0.703 | 0.391 | 20 | 0.004 | 0.382 | 0.176 | 35 | 0.0012 | 0.445 | 0.253 | 21 | 0.0007 |
| 11 | 25 |  | 100.0 |  | 307 | 0.03 | 31 | P3-100-10-05-16 | 0.0 |  | 14.681 | 7.630 | -122 | 0.520 | 21.325 | 11.170 | -90 | 0.524 | 36.006 | 18.800 | -212 | 0.522 |
| 10 | 25 |  | 100.0 |  | 307 | 0.01 |  | P3-100-20-05-16 | 0.0 | 5 | 7.706 | 4.014 | -54 | 0.521 | 27.344 | 14.284 | -153 | 0.522 | 35.050 | 18.299 | -208 | 0.522 |
| 14 | 25 |  | 100.0 |  | 307 | 0.05 | 21 | P3-100-03-06-16 | 0.2 |  | n.v. | n.v. | n.v. | n.v. | 35.233 | 18.403 | -200 | 0.522 | n.v. | n.v. | n.v. | n.v. |
|  |  |  |  |  |  | **Av.** 0.03 |  |  | **0.1** |  | **11.194** | **5.822** | **-88** | **0.520** | **27.968** | **14.619** | **-148** | **0.523** | **35.528** | **18.549** | **-210** | **0.522** |
|  |  |  |  |  |  | **SD** 0.02 |  |  | 0.1 |  | 4.932 | 2.557 | 48 | 0.001 | 6.975 | 3.628 | 55 | 0.0008 | 0.676 | 0.354 | 3 | 0.0000 |
|  |  |  |  |  |  |  |  | Av.(a) |  |  |  |  |  | 0.519 |  |  |  | 0.522 |  |  |  |  |
|  |  |  |  |  |  |  |  | SD (a) |  |  |  |  |  | 0.002 |  |  |  | 0.001 |  |  |  |  |
|  |  |  |  |  |  |  |  |  |  |  |  |  | λ=0.518 |  |  |  |  |  |  |  | λ=0.515 |  |

[revised manuscript text omitted]

a)

| | Laser analyzer Picarro L2140i (Ecotron) | | | IRMS MAT 253 (LSCE) | | | Difference laser analyzer/IRMS | | |
|---|---|---|---|---|---|---|---|---|---|
| | $\delta^{18}O$ | $\delta^{17}O$ | [17]O-excess | $\delta^{18}O$ | $\delta^{17}O$ | [17]O-excess | $\delta^{18}O$ | $\delta^{17}O$ | [17]O-excess |
| | ‰ | ‰ | per meg | ‰ | ‰ | per meg | ‰ | ‰ | per meg |
| GIENS-1 | -0.13 | -0.07 | 1.11 | -0.26 | -0.14 | -5.30 | 0.13 | 0.08 | 6.40 |
| ECO-1 | -5.68 | -2.97 | 29.13 | -5.61 | -2.94 | 28.10 | -0.07 | -0.03 | 1.04 |
| ICEBERG-1 | -26.88 | -14.25 | 36.09 | -27.13 | -14.38 | 35.43 | 0.24 | 0.13 | 0.66 |

b)

[revised manuscript text omitted]

Stomatal conductance: gs ranges from 0.1 to 0.5 in investigated C3 grasses is lower than 0.2 in C4 grasses. Cf Ocheltree et al., 2012. Here gs is calculated according to Liet al., 2017.

Boundary layer cond: 0.2 to 3 in Li et al., 2017

---

## Author Response (AR1)

**Point-by-point reply to the comments from Reviewers 1 and 2**

We would like to thank **reviewer #1** for his/her constructive comments. The points of concern are addressed below. In the revised draft found below, changes are highlighted in grey. Please note that in this revised draft, most recent references were added regarding the state of the art on biomarker isotope composition (section 1), the way RH is taken into account in GCMs (section 1) and the xylem water isotope signature (section 4.2) (Lehmann et al., 2018; Rach et al., 2017; Stevens et al., 2017).

**Major points**

- *Since there have been different measurement techniques used to determine the oxygen isotopes, it would be worthwhile in this context to report the comparability of the results mentioned in an additional table (H2O on Picarro L2140i and O2 converted from H2O on Delta V mass spectrometer) as well as the measurements done on the Picarro micro combustion module (MCM) in comparison with direct water measurements.*

In agreement with this comment, two tables were added in revised supplementary material: Table S1 **a)** Comparison between IRMS (4 replicates, SD of 0.015‰, 0.010‰ and 5 per meg for $\delta^{17}O$, $\delta^{18}O$ and $^{17}O$-excess respectively) and laser analyser (12 replicates, SD displayed) measurements of internal water standards. **b)** Measurements of some soil water and irrigation samples using the Picarro L2140i laser analyser without and with the microcombustion module (MCM). SD for standard deviation.

The other tables in the supplementary material are re-numbered accordingly.

In the revised draft, we wrongly mentioned the use of Delta V when a MAT 253 was actually used for IRMS measurements of the leaf water samples. The associated method is also corrected (from L324): The oxygen was then trapped in a tube immersed in liquid helium before being analyzed by dual inlet IRMS (ThermoQuest Finnigan MAT 253 mass spectrometer) against a reference oxygen gas. All measurements were run against a working $O_2$ standard calibrated against VSMOW. The resulting precisions (2 runs of 24 dual inlet measurements) were 0.015 ‰ for $\delta^{17}O$, 0.010 ‰ for and $\delta^{18}O$ and 5 per meg for $^{17}O$-excess.

- *How was the difference in $\Delta^{17}O$ between Phyto and LW calculated since from Fig. 1a, I am not able to obtain Fig. 1c for this difference? Please check it. This is also in line with the slopes of LW and Phyto vs. RH being different.*

Errors were made in the calculations of $^{17}O$-excess$_e$ and $\theta$. This is now corrected in the figures, tables and text. Calculations of $\delta'^{18}O$, $\delta'^{17}O$, $\Delta'^{18}O$ $^{17}O$-excess$_e$, $\theta$ and $\lambda$ are now detailed in the introduction section (L106). The text has been changed accordingly. The corrected data are close to the initial data and the interpretation of the data remains unchanged.

- *The comparison of the field with the lab results are critical (line 410 to 417), since there is no reason given why we should take the RH only for those months with a limited precipitation. This is in particular important since the r2 values actually decreases when going from RH or RH15 to the range limited by precipitation. This requires further discussion. It is no argument to fit the field data to the lab data just based on a slope measured.*

There may be a misunderstanding here. As stated in section 2.2, RH-rd0>1 is the averaged RH monthly means for months with at least one day with precipitation higher than 0.1 mm. It was calculated as a proxy of RH during the wet months, likely those of the grass growing season, which explains why the relationship between $^{17}O$-excess$_{phyto}$ and RH is the closest to the growth chamber's one.

For further clarity we add in the revised draft (L234): "In addition, in order to get a proxy of RH during wet months, likely those of the grass growing season, averaged RH monthly means for months with at least one day with precipitation higher than 0.1 mm (RH-rd0>1) was calculated.".

And in section 3.2 (L414): "The relationship obtained between $^{17}$O-excess$_{Phyto}$ and RH-rd0>1 (i.e. RH of the wet months during which plant grow) is the closest to the one obtained between $^{17}$O-excess$_{phyto}$ and RH in the growth chambers (fig. 4b)."

- *A weak point is indeed that no water vapor measurements are performed, this is indeed a strong shortcoming because a Picarro L2140i was available for the study. Yet, the authors clearly pointed out the importance to include such measurements in future studies. Was the leaf water measured for dD? If yes, this may help you with the interpretation in that it helps to make reasonable assumptions for the water vapor values.*

We indeed tried to measure δD on leaf water using a lazer analyzer Picarro L2130i. Still, as already demonstrated by previous studies (e.g. Schmidt et al., 2012), because of optical interference, the values are most of the time erroneous. This could be checked by a comparison with the δ$^{18}$O values produced by fluorination-IRMS showing that half of the Picarro δD values were off. We thus refrain from interpreting the δD values obtained through this approach.

- *Triple isotope comparison of Phyto with RH: It would be nice to distinguish the LW values given in blue for the high RH values (80-100 %) compared to low RH values (40%). This would allow the reader to better follow fig. 6. You may also use ellipses for these clarifications. Same issue with the Phyto values given in red.*

In agreement with this comment, figure 6 has been redone to differentiate phytolith and plant water data at 80-100%RH, 60%RH and 40%RH.

**Minor points:**

- *Why do you clean it cryogenically for NF, is NF produced during the fluorination process? How much could it affect the 17O and therefore the d17O results?*

We assume that NF may be produced from the fluorination of residual organic N in phytoliths. However, it is also possible that the interference of the $^{4}N^{9}F$ ion on the masse 33 is negligible. As a matter of fact we could not detect any $^{4}N^{9}F_2$ (mass 52) on the mass-spectrometer ThermoQuest Finnigan Delta Plus when analysing either terrestrial or extraterrestrial materials. Some of our internal quartz and phytolith standards were analysed with and without an extra slush-step. They gave similar results ($^{17}$O-excess of Boulangé: -0.110±0.031, n=148 without slush step, -0.104±0.022, n=63 with slush step; $^{17}$O-excess of MSG60: -0.216±0.033, n=22 without slush step, -0.212±0.043, n=7 with slush step). However, by caution, and in order to follow the same O$_2$ extraction protocol when analyzing terrestrial quartz, olivine, garnet and phytolith, as well as extra-terrestrial samples, the CEREGE stable isotopes laboratory chose to keep the slush step. Further comparisons with and without the slush step on several phytolith samples would be necessary to finally decide if the slush step is useful or not.

The revised draft was modified in section 2.6.1 as follows: "The purified oxygen gas (O$_2$) was passed through a –114 °C slush to refreeze gases interfering with the mass 33 (e.g. NF), potentially produced during the fluorination of residual organic N, before being sent to the dual-inlet mass spectrometer (ThermoQuest Finnigan Delta Plus)."

- *You mentioned that you checked the temperature independencies for 18O and 17O up to 70°C. Please add more information on this issue, because this is important. How have you done it? Wouldn't it be worthwhile to show the experimental results that you have obtained in this paper?*

In agreement with this comment we give details in the revised draft as follows (section 2.3, L249): "It has been shown that up to a temperature of 70 °C the extraction has no effect on the δ$^{18}$O (Crespin et al., 2008). We verified that it did not have any effect on the $^{17}O$-excess either, using our internal standard MSG extracted at 60 and 70°C (Crespin et al., 2008). The obtained $^{17}O$-excess values were similar (-211 and -243 per meg, respectively) given our reproducibility of ±34 per meg (see section 2.6.1)."

The way our internal phytolith standard MSG was extracted from a mascareignite soil sample at different temperature has been described in details in Crespin et al. (2008) as stated in the manuscript.

- *There is a significant difference of one of the standard material used, i.e. San Carlos Olivine. Whereas Sharp et al. (2016) reported a normalized δ18O value of 5.3 ‰ and a 17O-excess value of -54 per meg your values were δ18O SC = 4.949 ± 0.219 ‰ and 17O-excessSC = -49 ± 24 per meg. Why this difference in δ18O? δ18O*

This point is now discussed in the revised draft as follows (section 2.6.1, L311): "As previously discussed in Suavet et al. (2010), a large scatter is often observed for SC olivine $\delta^{18}O$ and $\delta^{17}O$ values measured in a given laboratory or from a laboratory to another. This is probably attributable to the heterogeneity of the analyzed samples. At CEREGE, the internal standard of SC olivine is prepared from a number of millimetric crystals with possibly different oxygen isotope composition. The $\delta^{18}O$ and $\delta^{17}O$ values from Suavet et al. (2010), Tanaka and Nakamura (2013) Pack et al. (2016), Sharp et al. (2016) and the present study average 5.29 ± 0.23 (1 SD) ‰ and 2.72 ± 0.12 (1 SD) ‰, respectively. Nevertheless, despite the large SD on $^{18}O$ and $\delta^{17}O$ measurements, the SC olivine $^{17}O$-excess appears relatively constant (-71 ± 23 (1 SD)) per meg."

- *On line 317 you have used ppm to express 17O-excess whereas you have often used per meg, be consistent over the whole manuscript.*

Corrected

**Specific remarks:**

- *l. 289: were dehydrated…do you mean adsorbed water or interstitial water?*

Corrected: dehydrated and dehydroxylated

- *l. 345: Make sure the minus sign is attached to the number.*

Corrected.

- *l. 362 etc.: Make sure that only relevant digits are given for the measurements according to their uncertainty.*

Three digits of precision on $\delta^{18}O$ and $\delta^{17}O$ values are necessary as 17O-excess is expressed in per meg.

- *l. 366: I suggest changing….withdrawn from the data set… to ….excluded from further calculations…*

Corrected.

- *l. 388: delete 00 prior to the number 2.*

Corrected.

- *l. 538: add space after for*

Corrected.

- *l. 542f: One can expect **that** the isotope composition…*

Corrected.

- *Table 1: Explain P1-40-29-04-16 etc. in the table legend.*

Corrected.

- *Table 2: Legend not consistent with table.*

Corrected.

- *Fig. 1: add x-axis on the top as well for easier readability. Panel c) is not consistent for me since it should be the difference between the measurements shown under panel a). This is not correct for all points. There should be an increase in Phyto-LW. Am I wrong?*

Reviewer is right. This was corrected in the revised version. See answers to the major points. x-axis is added on top of fig. 1a in the revised draft.

- *Fig. 5: How relevant is this figure?*

Fig.5 is not essential but is relevant to discuss the impact of the vegetation source and of the proportion of the Globular granulate phytoliths (assumed to come from the non-transpiring secondary xylem of the wood) on the $^{17}O$-excess of phytoliths. This is discussed in section 4.2.

- *Fig. 6: explain the different slope and slope ratios used in the figure.*

For further clarity, this is now explained in caption of figure 6 in the revised draft. The associated paragraph in the text was rewritten accordingly (section 4.2, L516).

We would like to thank **reviewer #2** for his/her constructive comments. The points of concern are addressed below. In the revised draft, changes are highlighted in grey.

*The authors present the results of triple oxygen isotope measurements of plant silica. The aim of this study is introducing d17O and d18O of phytoliths as proxy for the relative humidity (RH). The authors conducted laboratory experiments with controlled irrigation water composition, temperature and relative humidity. Data show that the difference in D17O between irrigation water and phytolith changes with RH. That is expected as the kinetic fractionation becomes more important at lower RH. Because kinetic fractionation follows a slope 0.516 (and 0.528 was used as reference line for defining D17O), resultant D17O values of the phytolith change with RH.*

*The authors point out that the weakness of this study is the lack of vapor data. That s true and the study would certainly have benefited from such data. The results show that the phytoliths fall on a line about parallel to the water evaporation trend typical of leaf water.*

- *In figure, the lines should go through all data, including the RH = 100 points. If necessary, draw curves. There is no physical reason why the laws of nature stop operating at RH = 85. It is a continuum; possibly with gradually changing mechanisms above RH = 85. That should not be camouflaged in the figure.*

In agreement with this comment, modifications were made to figure 1 as well as in the text (section 3.1) of the revised draft.

- *For readers with a b&w printer only, different symbols would be appropriate to distinguish the different data.*

This was modified in the revised draft.

- *In figure 1, "17O-excess" (top, bottom) is not relative to VSMOW. It is, however, reported relative to a reference line with slope (0.528) and intercept (0). Delete "VSMOW". Also, the Δ 18O should not be reported relative to VSMOW; it's a difference between δ values; delete VSMOW here, too.*

Right. This was modified in the revised draft.

- *Line 386ff: Don't give numbers like 27.948±7.168 ! Give 28±7. Only report significant number of digits. See also line 405; never give more digits than the uncertainty allows. Change throughout the entire manuscript.*

The precision on $\delta^{17}O$ and $\delta^{18}O$ should indeed be given with only 2 digits. Still, as shown in Landais et al. (2006) for leaf water, the uncertainties on $\delta^{17}O$ and $\delta^{18}O$ are not independent so that the final uncertainty on $^{17}O$-excess should not be calculated from uncorrelated uncertainties on $\delta^{17}O$ and $\delta^{18}O$. This is the reason why, there is a need to keep the 3 digits to properly calculate the $^{17}O$-excess. For further clarity 2 digits are presented in the text and 3 digits are kept in tables.

- *My major criticism on this paper is that it is not evaluated how precise (+/- RH values) the approach is for the reconstruction of the RH.*

In agreement with this comment, this is now discussed in the revised draft (section 4.2, L535):

"Without taking into account the two outliers, the linear regression between RH-rd0>1 and $^{17}O$-excess$_{phyto}$ for a 95% confidence interval can be expressed as follows:

RH-rd0>1 = 0.14 ± 0.02 (S.E) x $^{17}O$-excess$_{phyto}$ + 100.5 ± 4.7 (S.E)                          Eq. 2

where $^{17}O$-excess$_{phyto}$ is expressed in per meg and RH in %, $r^2$ = 0.48, and p < 0.001. S.E. stands for standard error. The S.E. of the predicted RH-rd0>1value is ± 5.6%. However, the data scattering (fig. 4) call for assessing additional parameters that can contribute to changes in $^{17}$O-excess$_{Phyto}$, beside RH, before using the $^{17}$O-excess$_{phyto}$ for quantitative RH reconstruction.

"

- *Also, completely missing is a discussion on the heterogeneity of leaf water and the effect on the phytolith composition. Eventually, people will use fossil phytoliths for reconstructing past RH and they will not know from which part of the plant the samples come.*

This is now discussed in section 4.2 (L552): "In grasses, leaf water is expected to be more prone to evaporative enrichment than stem water, and inside the leaf itself, the heterogeneity of evaporative sites repartition and water movements can lead to a significant heterogeneity in the $\delta^{18}$O signatures of water and phytoliths (Cernusak et al., 2016; Helliker and Ehleringer, 2000; Webb and Longstaffe, 2002). However, soil top phytolith assemblages likely record several decades of annual bulk phytolith production and their isotope composition is expected to be an average. This would explain the consistency of the $^{17}$O-excess$_{Phyto}$ data obtained from bulk grass phytoliths from climate chambers and the bulk phytolith assemblages from natural vegetation. Further investigation on the extent of the heterogeneity of $^{17}$O-excess in water and phytoliths in mature grasses would help to verify this assumption."

- *After this assessment, come to a decision on whether this proxy works or not (for useful applications). The lack of a quantitative assessment of all uncertainties is a general problem of many proxies. With some corrections and such a quantitative discussion, the manuscript is surely worth being published in Biogeosciences.*

We agree with all the points of concern raised above and gave a value of standard error on reconstructed RH (see answers above, e.g. L540: "However, the data scattering (fig. 4) call for assessing additional parameters that can contribute to changes in $^{17}$O-excess$_{Phyto}$, beside RH, before using the $^{17}$O-excess$_{phyto}$ for quantitative RH reconstruction."). However, as written in the discussion and conclusion sections we would like to emphasize that this is a first step in the assessment of the proxy and that complementary calibration steps are required to bring us to an accurate quantitative proxy. These complementary assessments are in progress.

**Additional changes**

- L59, in the abstract, the sentence "However, other parameters such as changes in the triple isotope composition of the soil water or phytolith origin in the plant may come into play" now replaces "However, other parameters such as changes in the triple isotope composition of the soil water or phytolith origin in the grass tissue may come into play", to include tree wood phytoliths discussed in the discussion section.

[revised manuscript text omitted]

---

## Author Response (AR2)

**Answer to reviewer #2**

**Without the file with the reviewer annotations, we did our best to take into account the reviewer comments/advices as follows:**

- Symbols θ and λ should be used with more care. We suggested a scheme that now becomes more and more accepted (Pack & Herwartz, 2014) with θ being used for a particular physical process (equilibrium or kinetic) and λ for regression on groups of materials NOT related through a well-understood single process and for the slope of a chosen reference line (then with index RL).
**This is clarified accordingly in the introduction section (L 111-114), in the text (grey underlining), in tables and on fig. 6.**

- The δ17O scaling to SMOW is still not clear. The authors cite studies that are either outdated or even did not analyze SMOW water. Be more careful here. You're dealing with water - "rock" interaction, so rocks need to be really on SMOW scale (better SMOW-SLAP). That's not trivial but can be done better than it appears in the current version of the manuscript. I put on remarks.
**The 'vs VSMOW' was removed when dealing with $^{17}$O-excess in two places in the text. As we don't have any file with reviewer 2 remarks it is hard to understand where the text can be further improved.**

- Meteoric water lines are known to span a curve. Hence, a "line" should not be longer used! I attach a compilation of some recent meteoric water analyses (I guess there are even more published now):
**We now refer to the meteoric water trend. This is only to explain how the $^{17}$O-excess was defined. The paragraph has been modified as follows: "** It has additionally been shown that meteoric waters plot along a trend with a slope λ of 0.528 ± 0.001. The departure from this trend is conventionally called $^{17}$O-excess ($^{17}$O-excess = δ'$^{17}$O - 0.528 x δ'$^{18}$O) (Luz and Barkan, 2010). **"**

- The kinetic fractionation in association with diffusion of vapor into the free atmosphere seems not to be considered as cause for variations of Δ'17O. Only the evaporation in leaves is considered. I may be wrong and the effect on the composition of the rain water is outbalanced by the larger effect due to evaporation in the leaves. A short comment on that would be welcome.
**The formulation of the kinetic effect during evaporation of water from the leaf is directly linked to the coefficient of diffusion of water vapor molecules in the free atmosphere in the classical formulation of isotopic enrichment of leaf water from the Craig and Gordon equation adapted to leaf (e.g. Cernusak et al., 2016). This kinetic effect is the driver of the $^{17}$O-excess signal in leaves (Landais et al., 2006; Li, Levin et al., 2017) and hence, in phytoliths, as explained in the manuscript. We checked everywhere in the manuscript, when diffusion and kinetic effect are mentionned, that the formulation do not lead to any ambiguity. We thus do not understand what the reviewer means exactly with this comment. We do not understand either the reference to Δ'17O that is never used in the manuscript.**

- In a few instances, still too many digits or precision are given.
**One correction was made (L358). We did not find other occurrences.**